# Promoting the modal shift of freight from road to rail in China: An evolutionary game and simulation study

Shuai Liu[1]*, Guangzhi Jia[2]

**1** Postgraduate Department, China Academy of Railway Sciences, Beijing, China, **2** Scientific & Technical Information Research Institute, China Academy of Railway Sciences Corporation Limited, Beijing, China

* liushuai.rjxh@foxmail.com

## Abstract

In the context of China's road to rail policy, this study constructs a tripartite evolutionary game model to mathematically investigate the strategic interactions among local governments, railway transport enterprises, and shippers. By employing mathematical modeling and simulation analysis, we examine the dynamic evolution of each stakeholder's strategy under varying conditions and verify the stability of the model. The findings indicate that government subsidies have a substantial effect on promoting shippers' adoption of rail transport, although this effect weakens over time. Furthermore, improvements in service quality by railway enterprises can significantly increase the attractiveness of rail transport, with shippers demonstrating the capacity to swiftly adapt to changing market conditions. This research highlights that an initial phase of government subsidies, followed by their gradual withdrawal, can effectively facilitate the modal shift from road to rail by leveraging market mechanisms. This study provides theoretical insights and practical guidance for the formulation of effective transport policies, emphasizing the pivotal role of government incentives, the proactive engagement of railway enterprises, and the adaptive behavior of shippers in advancing sustainable freight transport.

## Introduction

Rail freight transport is widely recognized as an environmentally sustainable mode of transportation because of its advantages of lower energy consumption, greater capacity, and reduced pollution [1,2]. In contrast, while offering considerable flexibility and convenience, road freight transport also poses challenges related to high energy consumption, increased pollution, and road congestion [3]. Consequently, promoting a shift from road to rail for freight transport has become a crucial objective of transportation policies across many countries [4–6]. In line with this objective, the Outline of the 14th Five-Year Plan for National Economic and Social Development and the Long-Range Objectives through the Year 2035 of the People's Republic of China explicitly call for accelerating the shift of bulk goods and medium-to long-distance freight from road to rail.

Rail transport offers significant environmental and economic advantages, such as lower energy consumption, high capacity, and reduced pollution. However, rail transport also has

**Data availability statement:** All relevant data are within the manuscript.

**Funding:** The author(s) received no specific funding for this work.

limitations, such as insufficient flexibility and longer cargo turnover times, which hinder its ability to meet the needs of enterprises requiring fast, point-to-point transportation [7]. As a result, rail freight struggles to fully leverage its potential advantages, and its market share remains relatively small. To address this issue and enhance rail freight market competitiveness, numerous scholars have conducted studies and provided valuable insights. Wang et al. [8] emphasized the need to integrate rail transport with logistics services, highlighting that a comprehensive logistics chain approach, which involves not only addressing warehousing but also tackling vehicle marshalling, sorting, and delivery to achieve "door-to-door" logistics services, is essential. Zhang et al. [9] suggested that the railway sector consider establishing its own road logistics companies, integrating drayage transport into the rail transport network, and introducing competitive mechanisms to improve quality and cost management in drayage services.

To further understand and effectively promote the shift in freight transport modes, extensive studies have been conducted both domestically and internationally. Lin [10] noted that rail freight volume is influenced heavily by policy direction, with significant fluctuations driven by policy changes. Feng [11] argued that accelerating the implementation of transport structure adjustment policies and promoting the transformation from road to rail requires a combination of market dynamics and comprehensive governmental measures, including administrative, fiscal, and legal support. Rotaris et al. [12] suggested that a modal shift from road transport to combined road–rail transport could have a substantial positive environmental impact. Moreover, Wisetjindawat et al. [13] reported that although road transport has a large market share in long-distance freight, strategies are needed to promote a shift toward rail services. Shi et al. [14] reported that when average train speeds exceed 50 km/h or loading times are shorter than 1.5 days, the market share of rail transport exceeds 50%. Furthermore, Shi et al. [15] concluded that rail freight volume fluctuates in response to changes in the prices of other transport modes, whereas the rail rates themselves significantly influence rail freight demand, with higher rates tending to decrease freight volume.

The road to rail model is also a form of multimodal transport, where integrating rail as a component of multimodal logistics is considered a key trend for future development. Numerous scholars have conducted studies in this area. For example, Borecka et al. [16] designed an optimized alternative service plan for a given set of line closures, including passenger rerouting, adjusted train schedules, bus bridging services, and additional train services. Ai et al. [17] developed a multi-objective optimization model that considers shippers' needs, the importance of timing in multimodal transport, and the advantages of carbon reduction. Zhang et al. [18] proposed a multimodal transport route selection model centered on rail, incorporating time penalty costs and damage compensation costs, aiming for the lowest overall transportation cost while considering transport reliability and safety.

As the discussion of multimodal transport and its relevance to the road to rail strategy continues, it becomes important to consider how stakeholders adapt their strategies within evolving market conditions. Evolutionary game theory provides a valuable framework for understanding the strategic dynamics between key players. Evolutionary game theory considers the bounded rationality of multiple stakeholders involved in a game and focuses on how these stakeholders adapt and evolve their strategies in a dynamic environment [19]. Several studies have been conducted in related fields. For example, Feng et al. [20] analyzed the strategy evolution in the China–Europe Railway Express market under subsidy withdrawal conditions. Yang and Zhang [21] emphasized the role of fiscal and tax subsidies as regulatory tools used by the government in balancing transport capacity and demand. Li et al. [22] suggested that a gradual withdrawal mechanism for government subsidies can promote coordination. Zhang and Xu [23] analyzed the optimal strategies for competition and cooperation among

nodal cities by comparing the demand, profit, and social welfare of platform companies under three scenarios, i.e., perfect competition, proactive platform cooperation, and government-led cooperation, with numerical analysis. Chen et al. [24] developed a game theoretic model to analyze the competition between maritime and rail transport. Furthermore, Tamannaei et al. [25] constructed a game theoretic model to study the competition between road transport and multimodal road–rail systems under government intervention.

Numerical analysis methods are crucial for evolutionary game theory because they enable the precise simulation, validation, and exploration of complex evolutionary dynamics and strategy stability under diverse conditions. Cârdei et al. [26] employed numerical analysis methods to establish and solve mathematical models of heavy metal transfer, enabling the simulation-based analysis of environmental factors affecting plant growth and metal absorption while optimizing remediation strategies; Du et al. [27] employed numerical analysis methods to validate a three-party evolutionary game model based on evolutionary game theory, simulating the evolutionary paths of decision-making behaviors among governments, contractors, and the public under varying levels of regulatory intensity, incentive mechanisms, and penalties and revealing the stable strategies and key influencing factors in construction and demolition waste management. Zou et al. [28] employed numerical analysis methods to validate a three-party evolutionary game model of an enterprise green innovation ecosystem on the basis of evolutionary game theory. Through simulation, the above authors analyzed the behavioral evolution paths of core enterprises and upstream and downstream enterprises under varying levels of subsidies, cooperation intensity, and initial willingness, revealing the evolutionary patterns and influencing factors of system vulnerability.

This study seeks to address the fundamental question of how local governments, railway transport enterprises, and shippers dynamically adjust their strategies under the road to rail policy in China and what key factors influence the stability and effectiveness of these strategic interactions. The primary objective of this work is to develop a tripartite evolutionary game model to systematically analyze the dynamic evolution of stakeholders' strategies across varying policy and market conditions while identifying optimal intervention points for government subsidies and service quality improvements. Driven by the urgent need to optimize China's freight transportation structure, this research aims to address persistent challenges such as policy inertia, stakeholder misalignment, and inadequate coordination mechanisms. Although subsidies are pivotal in initiating behavioral changes, the timing and mechanisms for their gradual withdrawal remain underexplored, as does the long-term sustainability of improvements in railway service quality. Existing studies rely predominantly on static analyses, often neglecting the temporal dynamics and interdependencies that define stakeholder interactions. To address this gap, this study integrates evolutionary game theory with numerical simulation analysis to capture the temporal evolution of strategic adjustments and identify stable equilibrium conditions. This approach reveals asymmetries in strategy adaptation speeds among stakeholders and emphasizes the necessity of synchronized policy interventions. Furthermore, this work underscores the critical role of service quality enhancements in sustaining long-term behavioral shifts among shippers. By bridging the gap between theoretical insights and practical policy recommendations, this study not only advances the academic understanding of stakeholder dynamics in the road to rail transition in China but also provides actionable strategies to guide policy implementation, ensuring the long-term sustainability and effectiveness of this critical transportation shift.

## Evolutionary game theoretic model

### Problem description

The tripartite evolutionary game method effectively simulates interactions among the government, railway transport enterprises, and shippers, capturing the strategy evolution of each

party under bounded rationality and identifying the long-term stable state of the system via dynamic analysis. This method allows for the evaluation of different policies (such as government subsidies and service improvements by railway transport enterprises), aiding in the formulation of more practical policy combinations and promoting the shift in the transport structure from road to rail. The logical relationships among the tripartite evolutionary game participants constructed in this study are illustrated in Fig 1.

## Model assumptions and notations

To facilitate the analysis of the constructed tripartite evolutionary game model, including the evolutionary strategies of each game participant, the interrelationships among the participants, and the stability analysis of strategy equilibrium points, the following assumptions are made:

**Assumption 1:** The loparacal government, railway transport enterprises and shippers are considered participants in the tripartite evolutionary game, each acting as a bounded rational agent aiming to maximize its own interests. Participants face information asymmetry, and their strategies evolve toward an optimal strategy over time.

**Assumption 2:** The strategy set of the local government is $S_\tau = \{ S_{\tau_1}, S_{\tau_2} \}$, where $S_{\tau_1}$ represents the strategy of providing subsidies to shippers and $S_{\tau_2}$ represents the strategy of not providing subsidies. The strategy set of the railway transport enterprises is $S_\omega = \{ S_{\omega_1}, S_{\omega_2} \}$, where $S_{\omega_1}$ represents the strategy of actively cooperating and $S_{\omega_2}$ represents the strategy of maintaining the status quo. The strategy set of the shippers is $S_\phi = \{ S_{\phi_1}, S_{\phi_2} \}$, where $S_{\phi_1}$ represents actively choosing rail transport as the mode of freight transport and $S_{\phi_2}$ represents actively choosing road transport as the mode of freight transport.

**Assumption 3:** The probability of the local government choosing strategy $S_{\tau_1}$, providing subsidies to shippers, is $x$, and the probability of it choosing strategy $S_{\tau_2}$, not providing subsidies, is $1-x$, where $x \in [0,1]$. The probability of railway transport enterprises choosing strategy $S_{\omega_1}$, actively cooperating, is $y$, and the probability of theme choosing strategy $S_{\omega_2}$, maintaining the status quo, is

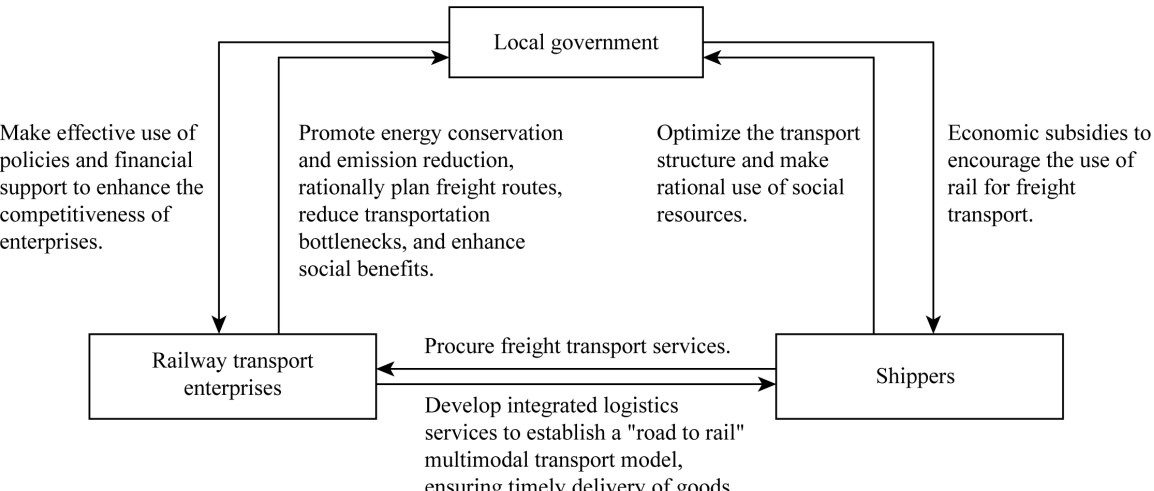

**Fig 1. Logical relationships among tripartite evolutionary game participants.**

1- $y$, where $y \in [0,1]$. The probability of shippers actively choosing rail as the mode of freight transport $S_{\phi_1}$ is $z$, whereas the probability of them choosing road transport $S_{\phi_2}$ is 1- $z$, where $z \in [0,1]$.

**Assumption 4:** From the perspective of the local government, the total economic, environmental, and social benefits derived from transporting one unit of freight by road are denoted as $G_{rh}$. The total economic, environmental, and social benefits derived from transporting one unit of freight by rail are denoted as $G_{rr}$. The environmental and economic benefits resulting from shifting freight transport from road to rail are represented by $G_t$. The investment made by the local government in building new road–rail intermodal facilities to promote the shift is denoted as $G_{ei}$. The subsidy provided by the local government to shippers choosing rail as their mode of transport is represented by $G_{es}$. The cost of addressing additional energy consumption and environmental issues caused by road transport compared with rail transport is denoted as $G_{ee}$.

The local government has a certain probability, $\delta$, where $\delta \in [0,1]$, of supervising the shippers receiving subsidies. The cost of supervision is denoted as $G_{cs}$. It is assumed that any noncompliant behavior by shippers in receiving subsidies will certainly be detected. If shippers apply for subsidies but fail to carry out the related freight transport as needed, then subsidy $G_{es}$ will be recovered and a penalty $G_f$ will be imposed.

**Assumption 5:** From the perspective of railway transport enterprises, the operating cost is denoted as $R_{ec}$, and the profit is denoted as $R_{rp}$. When freight transport shifts from road to rail, the additional cost incurred is represented by $R_{ec}^{'}$, and the additional profit obtained is represented by $R_{rp}^{'}$.

If railway transport enterprises fail to complete the construction of new road–rail intermodal facilities, then there is a certain probability, $\varepsilon$, where $\varepsilon \in [0,1]$, that owing to issues such as capacity constraints, the freight will not be delivered on time, resulting in compensation $R_f$ being paid to shippers.

**Assumption 6:** From the perspective of shippers, all goods can be transported by either rail or road, with the total quantity of goods being $Q$. Currently, the proportion of goods transported by rail is $Q_1$, and the proportion transported by road is $Q_2$, with the following relationship:

$$Q = Q_1 + Q_2 \tag{1}$$

After subsidies, if the total quantity of goods transported by rail is $Q_1^{'}$ and that transported by road is $Q_2^{'}$, then the relationship is as follows:

$$Q = Q_1^{'} + Q_2^{'} \tag{2}$$

For shippers, the total cost of transporting one unit of freight by rail is $S_{pr}$, whereas the cost of transporting one unit by road is $S_{ph}$. Compared with road transport, rail transport, when capacity is sufficient, can handle larger quantities of the same type of goods per shipment, resulting in savings in time and economic profit, represented by $S_e$.

The notations of the tripartite evolutionary game, which represent the strategic interactions among local governments, railway transport enterprises, and shippers, are summarized in Table 1.

**Table 1. Notations of the Tripartite Evolutionary Game.**

| Notations | Definition |
|---|---|
| $S_{i(i=\tau_1, \tau_2, \omega_1, \omega_2, \phi_1, \phi_2)}$ | Strategies of the local government, railway transport enterprises and shippers |
| $x\ y\ z$ | Probability of strategy selection by the local government, railway transport enterprises and shippers |
| $G_{rh}$ | Total economic, environmental, and social benefits derived from transporting one unit of freight by road |
| $G_{rr}$ | Total economic, environmental, and social benefits derived from transporting one unit of freight by rail |
| $G_t$ | Environmental and economic benefits resulting from shifting freight transport from road to rail |
| $G_{ei}$ | Investment made by the local government in building new road-rail intermodal facilities to promote the shift |
| $G_{es}$ | Subsidy provided by the local government to shippers choosing rail as their mode of transport |
| $G_{ee}$ | Cost of addressing additional energy consumption and environmental issues caused by road transport compared to rail transport |
| $\delta$ | Probability of the local government supervising shippers receiving subsidies |
| $G_{cs}$ | Cost incurred by the local government for supervising shippers receiving subsidies |
| $G_f$ | Penalty imposed on shippers for noncompliant behavior in receiving subsidies if they fail to carry out the related freight transport as required |
| $R_{ec}$ | Operating cost of railway transport enterprises |
| $R_{ec}'$ | Additional operating cost incurred after shifting freight transport from road to rail |
| $R_{rp}$ | Profit of railway transport enterprises |
| $R_{rp}'$ | Additional profit gained after shifting freight transport from road to rail |
| $\varepsilon$ | Probability that if railway transport enterprises fail to complete the construction of new road-rail intermodal facilities, then they may fail to deliver freight on time, resulting in compensation paid to shippers |
| $R_f$ | Compensation paid to shippers due to their failure to deliver freight on time as a result of capacity constraints |
| $Q$ | Total quantity of goods that can be transported by either rail or road |
| $Q_1\ Q_2$ | Proportion of goods transported by rail/road |
| $Q_1'\ Q_2'$ | Proportion of goods transported by rail/road after receiving subsidies |
| $S_{pr}\ S_{ph}$ | Total cost of transporting one unit of freight by rail/road |
| $S_e$ | Savings in time and economic profit resulting from rail transport compared to road transport, especially when capacity is sufficient and larger quantities can be shipped |

## Modeling framework

On the basis of the above assumption, the payoff matrix for the local government, railway transport enterprises, and shippers is constructed, as shown in Table 2.

## Model solution and analysis

Before delving into the evolutionary game model and simulation analysis, it is essential to reference some classic representative literature in the fields of game theory and simulation analysis. These works not only illustrate the fundamental principles of evolutionary game algorithms but also provide a theoretical framework and developmental direction for current research.

Smith [29] elaborated on the application of game theory to evolutionary biology, introducing the core concept of the evolutionary stable strategy (ESS). He emphasized that in conflict-of-interest scenarios, the optimal strategy depends on the strategies adopted by others. Additionally, the above study introduced the mathematical framework of replicator dynamics, exploring the conditions for strategy stability and its application scenarios in depth and laying the theoretical foundation for the subsequent development of evolutionary game theory. Taylor and Jonker [30] developed a mathematical model to describe the dynamic

**Table 2. Payoff Matrix.**

| Tripartite Evolutionary Game Strategy | $S_{\lambda_1}$: Local government provides subsidies to shippers | | $S_{\lambda_2}$: Local government does not provide subsidies to shippers | |
|---|---|---|---|---|
| | $S_{\omega_1}$: Railway transport enterprises actively cooperate | $S_{\omega_2}$: Railway transport enterprises maintain the status quo | $S_{\omega_1}$: Railway transport enterprises actively cooperate | $S_{\omega_2}$: Railway transport enterprises maintain the status quo |
| $S_{\phi_1}$: Shippers actively choose rail transport | $G_{rr}Q_1' + G_{rh}Q_2' + G_t$ $-G_{ei} - G_{es}$ $-G_{cs} + \delta(G_{es} + G_f)$ | $G_{rr}Q_1' + G_{rh}Q_2' + G_t$ $-G_{es} - G_{cs}$ $+\delta(G_{es} + G_f)$ | $G_{rr}Q_1' + G_{rh}Q_2'$ $+G_t - G_{ei}$ | $G_{rr}Q_1' + G_{rh}Q_2' + G_t$ |
| | $R_{rp} - R_{ec} + R_{rp}'$ $-R_{ec}' + G_{ei}$ | $R_{rp} - R_{ec} + R_{rp}'$ $-R_{ec}' - \varepsilon R_f$ | $R_{rp} - R_{ec} + R_{rp}'$ $-R_{ec}' + G_{ei}$ | $R_{rp} - R_{ec} - \varepsilon R_f$ |
| | $G_{es} - Q_1'S_{pr} - Q_2'S_{ph}$ $-\delta(G_{es} + G_f) + S_e$ | $G_{es} - Q_1'S_{pr} - Q_2'S_{ph}$ $-\delta(G_{es} + G_f) + \varepsilon R_f$ | $-Q_1'S_{pr} - Q_2'S_{ph}$ | $-Q_1'S_{pr} - Q_2'S_{ph} + \varepsilon R_f$ |
| $S_{\phi_2}$: Shippers actively choose road transport | $G_{rr}Q_1 + G_{rh}Q_2$ $-G_{ei} - G_{ee} - G_{cs}$ | $G_{rr}Q_1 + G_{rh}Q_2$ $-G_{ee} - G_{cs}$ | $G_{rr}Q_1 + G_{rh}Q_2$ $-G_{ei} - G_{ee}$ | $G_{rr}Q_1 + G_{rh}Q_2$ $-G_{ee}$ |
| | $R_{rp} - R_{ec} + G_{ei}$ | $R_{rp} - R_{ec} - \varepsilon R_f$ | $R_{rp} - R_{ec} + G_{ei}$ | $R_{rp} - R_{ec} - \varepsilon R_f$ |
| | $-Q_1 S_{pr} - Q_2 S_{ph}$ | $-Q_1 S_{pr} - Q_2 S_{ph} + \varepsilon R_f$ | $-Q_1 S_{pr} - Q_2 S_{ph}$ | $-Q_1 S_{pr} - Q_2 S_{ph} + \varepsilon R_f$ |

processes of evolutionary games using nonlinear first-order differential equations for the continuous case and nonlinear difference equations for the discrete case. The central conclusion of the study was that under certain nondegeneracy conditions, an ESS is always stable in continuous dynamic systems but not necessarily stable in discrete dynamic systems. such research provides an important theoretical framework for evolutionary game theory and lays a solid foundation for subsequent studies on game dynamics.

## Stability analysis of local government strategies

By analyzing the stability of local government strategies, the evolutionary process of government behavior under different subsidy policies can be observed. Regardless of whether subsidies are provided, the government's strategic choices significantly influence the behavior of shippers and railway transport enterprises during the game process, which not only provides a basis for the formulation of government policies but also reveals their key role in promoting the road to rail transition. The expected payoff and calculation of the average expected payoff for the local government are as follows:

The expected payoff $E_{\tau_1}$ for the local government when choosing the strategy $S_{\tau_1}$ of providing subsidies to shippers is as follows:

$$
\begin{aligned}
E_{\tau_1} = \ & yz[G_{rr}Q_1' + G_{rh}Q_2' + G_t - G_{ei} - G_{es} - G_{cs} + \delta(G_{es} + G_f)] \\
& + y(1-z)(G_{rr}Q_1 + G_{rh}Q_2 - G_{ei} - G_{ee} - G_{cs}) \\
& + (1-y)z[G_{rr}Q_1' + G_{rh}Q_2' + G_t - G_{es} - G_{cs} + \delta(G_{es} + G_f)] \\
& + (1-y)(1-z)(G_{rr}Q_1 + G_{rh}Q_2 - G_{ee} - G_{cs})
\end{aligned}
\tag{3}
$$

The expected payoff $E_{\tau_2}$ for the local government when choosing the strategy $S_{\tau_2}$ of not providing subsidies to shippers is as follows:

$$
\begin{aligned}
E_{\tau_2} = \ & yz(G_{rr}Q_1' + G_{rh}Q_2' + G_t - G_{ei}) + y(1-z)(G_{rr}Q_1 + G_{rh}Q_2 - G_{ei} - G_{ee}) \\
& + (1-y)z(G_{rr}Q_1' + G_{rh}Q_2') + (1-y)(1-z)(G_{rr}Q_1 + G_{rh}Q_2 - G_{ee})
\end{aligned}
\tag{4}
$$

The average payoff for the local government, denoted as $\bar{E}_\tau$, is as follows:

$$\bar{E}_\tau = xE_{\tau_1} + (1-x)E_{\tau_2} \tag{5}$$

The replicator dynamic equation for the local government is as follows:

$$F(x) = \frac{dx}{dt} = x\left(E_{\tau_1} - \bar{E}_\tau\right) = x\left[E_{\tau_1} - xE_{\tau_1} - (1-x)E_{\tau_2}\right] = x(1-x)\left(E_{\tau_1} - E_{\tau_2}\right) \tag{6}$$

The first partial derivative of $F(x)$ with respect to $x$ is then obtained, and Equations (3) and (4) are substituted to derive the following relationship:

$$\frac{dF(x)}{dx} = (1-2x)\left(E_{\tau_1} - E_{\tau_2}\right) = (1-2x)\left\{-G_{cs} + z\left[(\delta-1)G_{es} + \delta G_f + (1-y)G_t\right]\right\} \tag{7}$$

By letting $-G_{cs} + z\left[(\delta-1)G_{es} + \delta G_f + (1-y)G_t\right] = 0$ and solving for $z^*$, we obtain the following:

$$z^* = \frac{G_{cs}}{-yG_t - G_{es} + \delta G_{es} + \delta G_f + G_t} \tag{8}$$

When $z = z^*$, $F(x) \equiv 0$, and all values of $x$ are in a stable state. When $0 \leq z \leq z^* \leq 1$, $\frac{dF(x)}{dx}\Big|_{x=0} > 0$ and $\frac{dF(x)}{dx}\Big|_{x=1} < 0$, indicating that the equilibrium point is $x = 1$. Similarly, when $0 \leq z^* \leq z \leq 1$, $\frac{dF(x)}{dx}\Big|_{x=0} < 0$ and $\frac{dF(x)}{dx}\Big|_{x=1} > 0$, indicating that the equilibrium point is $x = 0$. A phase diagram of the evolutionary behavior of government strategies is shown in Fig 2.

The stability analysis of railway transport enterprises' strategies reveals their behavioral evolution under different market environments and policy conditions. When market demand increases or when the government provides policy incentives, railway transport enterprises gradually tend to actively cooperate by optimizing service quality to increase competitiveness. Although this proactive market response may take a longer time to yield significant effects, once enterprises increase investment and improve service capabilities, shippers' preferences also shift toward rail transport. The expected payoff and the calculation of the average expected payoff for railway transport enterprises are as follows.

The expected payoff $E_{\omega_1}$ for railway transport enterprises when choosing strategy $S_{\omega_1}$ for actively cooperating is as follows:

$$\begin{aligned}
E_{\omega_1} = &\, xz\left(R_{rp} - R_{ec} + R'_{rp} - R'_{ec} + G_{ei}\right) + x(1-z)\left(R_{rp} - R_{ec} + G_{ei}\right) \\
&+ (1-x)z\left(R_{rp} - R_{ec} + R'_{rp} - R'_{ec} + G_{ei}\right) \\
&+ (1-x)(1-z)\left(R_{rp} - R_{ec} + G_{ei}\right)
\end{aligned} \tag{9}$$

The expected payoff $E_{\omega_2}$ for railway transport enterprises when choosing strategy $S_{\omega_2}$ for maintaining the status quo is as follows:

$$\begin{aligned}
E_{\omega_2} = &\, xz\left(R_{rp} - R_{ec} + R'_{rp} - R'_{ec} - \varepsilon R_f\right) + x(1-z)\left(R_{rp} - R_{ec} - \varepsilon R_f\right) \\
&+ (1-x)z\left(R_{rp} - R_{ec} - \varepsilon R_f\right) \\
&+ (1-x)(1-z)\left(R_{rp} - R_{ec} - \varepsilon R_f\right)
\end{aligned} \tag{10}$$

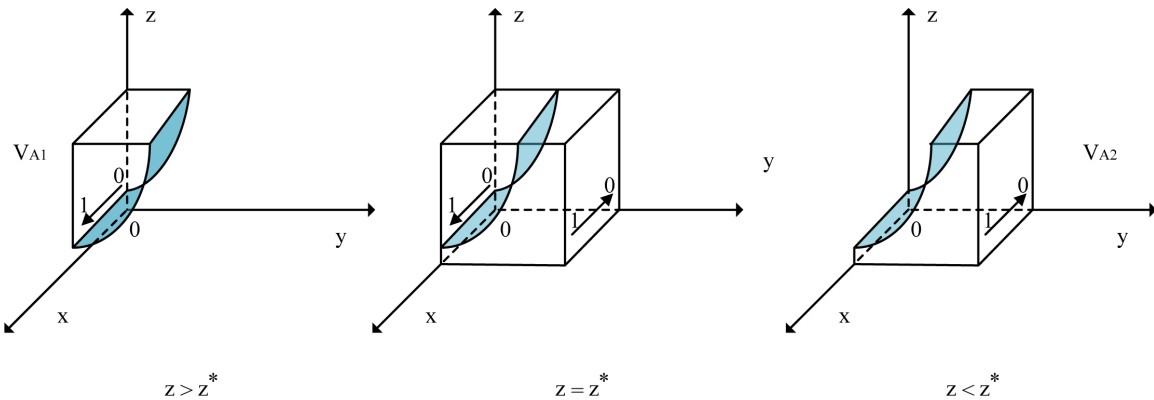

**Fig 2. Phase Diagram of Local Government Strategy Evolution.** Notes: "$V_{A_1}$" designates the strategy space applicable when $z > z^*$, and the arrow extending from "0" to "1" indicates that within this strategy space, the evolutionary trajectory of x progresses from "0" to "1"; "$V_{A_2}$" designates the strategy space applicable when $z < z^*$, and the arrow extending from "1" to "0" indicates that within this strategy space, the evolutionary trajectory of x progresses from "1" to "0"; and the shaded area in the figure represents the cross-sectional view of the strategy space under the condition that $z = z^*$.

The average payoff for railway transport enterprises, denoted as $\bar{E}_\omega$, is as follows:

$$\bar{E}_\omega = yE_{\omega_1} + (1-y)E_{\omega_2} \tag{11}$$

The replicator dynamic equation for railway transport enterprises is as follows:

$$F(y) = \frac{dy}{dt} = y\left(E_{\omega_1} - \bar{E}_\omega\right) = y\left[E_{\omega_1} - yE_{\omega_1} - (1-y)E_{\omega_2}\right] = y(1-y)\left(E_{\omega_1} - E_{\omega_2}\right) \tag{12}$$

The first partial derivative of $F(y)$ with respect to $y$ is then obtained, and Equations (9) and (10) are substituted to derive the following relationship:

$$\frac{dF(y)}{dy} = (1-2y)\left(E_{\omega_1} - E_{\omega_2}\right) = (1-2y)\left[G_{ei} + \varepsilon R_f + (1-x)z\left(R'_{rp} - R'_{ec}\right)\right] \tag{13}$$

By letting $G_{ei} + \varepsilon R_f + (1-x)z\left(R'_{rp} - R'_{ec}\right) = 0$ and solving for $x^*$, we obtain the following:

$$x^* = 1 + \frac{G_{ei} + \varepsilon R_f}{z\left(R'_{rp} - R'_{ec}\right)} \tag{14}$$

When $x = x^*$, $F(y) \equiv 0$, and all values of $y$ are in a stable state. When $0 \leq x \leq x^* \leq 1$, $\frac{dF(y)}{dy}|_{y=0} < 0$ and $\frac{dF(y)}{dy}|_{y=1} > 0$, indicating that the equilibrium point is $y = 0$. Similarly, when $0 \leq x^* \leq x \leq 1$, $\frac{dF(y)}{dy}|_{y=0} > 0$ and $\frac{dF(y)}{dy}|_{y=1} < 0$, indicating that the equilibrium point is $y = 1$. A phase diagram of the evolutionary behavior of railway transport enterprise strategies is shown in Fig 3.

## Stability analysis of shippers' strategies

As key participants in the game, the stability analysis of the strategies of shippers plays an important role in overall system stability. Shippers' strategy choices not only directly affect the proportion of rail and road transport but also have a feedback effect on the strategic

adjustments of local government and railway enterprises. The expected payoff and the calculation of the average expected payoff for shippers are presented below.

The expected payoff $E_{\phi_1}$ for shippers when choosing strategy $S_{\phi_1}$ of actively choosing rail transport is as follows:

$$E_{\phi_1} = xy\left[G_{es} - Q_1'S_{pr} - Q_2'S_{ph} - \delta(G_{es} + G_f) + S_e\right] + (1-x)y\left[G_{es} - Q_1'S_{pr} - Q_2'S_{ph} - \delta(G_{es} + G_f) + \varepsilon R_f\right] \\ + x(1-y)\left(-Q_1'S_{pr} - Q_2'S_{ph}\right) \\ + (1-x)(1-y)\left(-Q_1'S_{pr} - Q_2'S_{ph} + \varepsilon R_f\right) \tag{15}$$

The expected payoff $E_{\phi_2}$ for shippers when choosing strategy $S_{\phi_2}$ of actively choosing road transport is as follows:

$$E_{\phi_2} = xy\left(-Q_1 S_{pr} - Q_2 S_{ph}\right) + (1-x)y\left(-Q_1 S_{pr} - Q_2 S_{ph} + \varepsilon R_f\right) \\ + x(1-y)\left(-Q_1 S_{pr} - Q_2 S_{ph}\right) + (1-x)(1-y)\left(-Q_1 S_{pr} - Q_2 S_{ph} + \varepsilon R_f\right) \tag{16}$$

The average payoff for shippers, denoted as $\overline{E}_\phi$, is as follows:

$$\overline{E}_\phi = zE_{\phi_1} + (1-z)E_{\phi_2} \tag{17}$$

The replicator dynamic equation for shippers is as follows:

$$F(z) = \frac{dz}{dt} = z\left(E_{\phi_1} - \overline{E}_\phi\right) = z\left[E_{\phi_1} - zE_{\phi_1} - (1-z)E_{\phi_2}\right] = z(1-z)\left(E_{\phi_1} - E_{\phi_2}\right) \tag{18}$$

The first partial derivative of $F(z)$ with respect to $z$ is then obtained, and Equations (9) and (10) are substituted to derive the following relationship:

$$\frac{dF(z)}{dt} = (1-2z)\left(E_{\phi_1} - E_{\phi_2}\right) = (1-2z)\left[(1-\delta)yG_{es} - \delta yG_f + xyS_e + Q_2 S_{ph} - Q_2'S_{ph} + Q_1 S_{pr} - Q_1'S_{pr}\right] \tag{19}$$

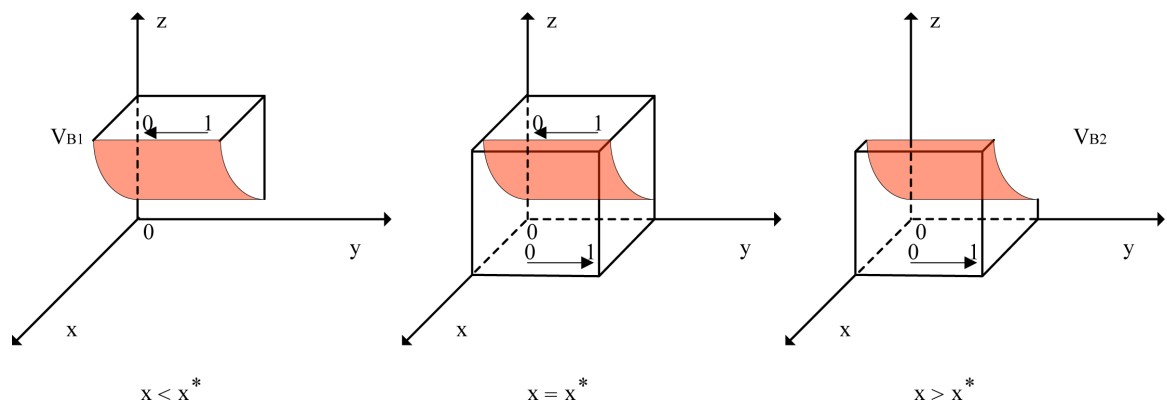

**Fig 3. Phase Diagram of Railway Transport Enterprises' Behavior Evolution.** Note: "$V_{B1}$" designates the strategy space applicable when $x < x^*$, and the arrow extending from "1" to "0" indicates that within this strategy space, the evolutionary trajectory of y progresses from "1" to "0"; "$V_{B2}$" designates the strategy space applicable when $x > x^*$, and the arrow extending from "0" to "1" indicates that within this strategy space, the evolutionary trajectory of y progresses from "0" to "1"; and the shaded area in the figure represents the cross-sectional view of the strategy space under the condition that $x = x^*$.

By letting $(1-\delta)yG_{es} - \delta yG_f + xyS_e + Q_2S_{ph} - Q_2'S_{ph} + Q_1S_{pr} - Q_1'S_{pr} = 0$ and solving for $y^*$, we obtain the following:

$$y^* = \frac{Q_2S_{ph} - Q_2'S_{ph} + Q_1S_{pr} - Q_1'S_{pr}}{-G_{es} + \delta G_{es} + \delta G_f - xS_e} \quad (20)$$

When $y = y^*$, $F(z) \equiv 0$, and all values of $z$ are in a stable state. When $0 \le y \le y^* \le 1$, $\frac{dF(z)}{dz}\big|_{z=0} > 0$ and $\frac{dF(z)}{dz}\big|_{z=1} < 0$, indicating that the equilibrium point is $z = 1$. Similarly, when $0 \le y^* \le y \le 1$, $\frac{dF(z)}{dz}\big|_{z=0} < 0$ and $\frac{dF(z)}{dz}\big|_{z=1} > 0$, indicating that the equilibrium point is $z = 0$. A phase diagram of the evolutionary behavior of shipper strategies is shown in Fig 4.

### Stability analysis of equilibrium points in the tripartite evolutionary game system

The above three subsections analyze the strategy evolution processes of the local government, railway transport enterprises, and shippers as individual participants. However, the tripartite evolutionary game system is the result of the combined evolution of strategies from all three parties. Therefore, by combining the dynamic equations of the local government, railway transport enterprises, and shippers, we obtain the following:

$$\begin{cases} F(x) = x(1-x)\{-G_{cs} + z[(\delta-1)G_{es} + \delta G_f + (1-y)G_t]\} \\ F(y) = y(1-y)[G_{ei} + \varepsilon R_f + (1-x)z(R_{rp}' - R_{ec}')] \\ F(z) = z(1-z)[(1-\delta)yG_{es} - \delta yG_f + xyS_e + Q_2S_{ph} - Q_2'S_{ph} + Q_1S_{pr} - Q_1'S_{pr}] \end{cases} \quad (21)$$

The Jacobian matrix of the tripartite evolutionary game system is as follows:

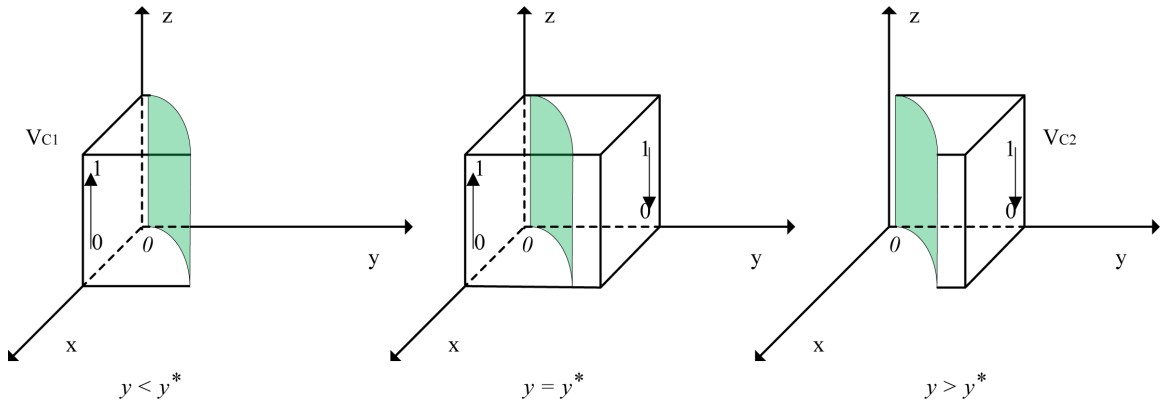

**Fig 4. Phase Diagram of Shippers' Strategy Evolution.** Note: "VC1" designates the strategy space applicable when y < y*, and the arrow extending from "0" to "1" indicates that within this strategy space, the evolutionary trajectory of z progresses from "0" to "1"; "VC2" designates the strategy space applicable when y > y*, and the arrow extending from "1" to "0" indicates that within this strategy space, the evolutionary trajectory of z progresses from "1" to "0"; and the shaded area in the figure represents the cross-sectional view of the strategy space under the condition that y = y*.

$$J = \begin{bmatrix} J_1 & J_2 & J_3 \\ J_4 & J_5 & J_6 \\ J_7 & J_8 & J_9 \end{bmatrix} = \begin{vmatrix} \dfrac{\partial F(x)}{\partial x} & \dfrac{\partial F(x)}{\partial y} & \dfrac{\partial F(x)}{\partial z} \\ \dfrac{\partial F(y)}{\partial x} & \dfrac{\partial F(y)}{\partial y} & \dfrac{\partial F(y)}{\partial z} \\ \dfrac{\partial F(z)}{\partial x} & \dfrac{\partial F(z)}{\partial y} & \dfrac{\partial F(z)}{\partial z} \end{vmatrix} \tag{22}$$

where

$$\frac{\partial F(x)}{\partial x} = (1-2x)\left\{-G_{cs} + z\left[(-1+a)G_{es} + aG_f + G_t - yG_t\right]\right\} \tag{23}$$

$$\frac{\partial F(x)}{\partial y} = -(1-x)xzG_t \tag{24}$$

$$\frac{\partial F(x)}{\partial z} = (1-x)x\left[(-1+a)G_{es} + aG_f + G_t - yG_t\right] \tag{25}$$

$$\frac{\partial F(y)}{\partial x} = -(-1+y)yz\left(R'_{ec} - R'_{rp}\right) \tag{26}$$

$$\frac{\partial F(y)}{\partial y} = (1-2y)\left[G_{ei} + bR_f + (-1+x)z\left(R'_{ec} - R'_{rp}\right)\right] \tag{27}$$

$$\frac{\partial F(y)}{\partial z} = -(-1+x)(-1+y)y\left(R'_{ec} - R'_{rp}\right) \tag{28}$$

$$\frac{\partial F(z)}{\partial x} = y(1-z)zS_e \tag{29}$$

$$\frac{\partial F(z)}{\partial y} = (1-z)z\left[(1-a)G_{es} - aG_f + xS_e\right] \tag{30}$$

$$\frac{\partial F(z)}{\partial z} = (1-2z)\left[(y-ay)G_{es} - ayG_f + xyS_e + Q_2S_{ph} - Q_{22}S_{ph} + Q_1S_{pr}\right] \tag{31}$$

The pure strategy equilibrium points are $E_1(0,0,0)$, $E_2(0,0,1)$, $E_3(0,1,0)$, $E_4(1,0,0)$, $E_5(0,1,1)$, $E_6(1,0,1)$, $E_7(1,1,0)$, and $E_8(1,1,1)$. The mixed strategy equilibrium points are $E_9(x^*,0,0)$, $E_{10}(0,y^*,0)$, and $E_{11}(0,0,z^*)$ and represent a nonevolutionary stable strategy [31].

According to Lyapunov's first method (indirect method), a pure strategy equilibrium point is a stable strategy if all the eigenvalues of the constructed Jacobian matrix have negative real parts. If any eigenvalue has a positive real part, then the equilibrium point is either unstable or indeterminate. The eigenvalues of the Jacobian matrix for pure strategy equilibrium points are shown in Table 3.

**Scenario 1.** When $G_{ei} - R_{ec} + \varepsilon R_f + R'_{rp} < 0$, all the eigenvalues of the Jacobian matrix constructed at $E_2(0,0,1)$ have negative real parts, indicating a stable equilibrium point. The

corresponding evolutionary game strategy is $S_1 = \{$the local government does not provide subsidies to shippers, railway transport enterprises maintain the status quo, and shippers actively choose rail transport$\}$.

**Analysis:** This represents an ideal equilibrium state, where the local government does not rely on fiscal incentives to drive transport adjustments, railway transport enterprises achieve profit through maintaining their existing operations, and shippers make an active choice owing to the cost advantages or environmental benefits of rail transport. However, the sustainability of this equilibrium depends on changes in market conditions and the dynamic adjustments of each party's behavior. Without sufficient incentives and reforms, the enthusiasm of shippers may gradually diminish, thereby affecting the competitiveness of rail transport.

**Scenario 2.** When $G_{es} - \delta G_{es} - \delta G_f + Q_2 S_{ph} - Q_2' S_{ph} + Q_1 S_{pr} - Q_1' S_{pr} < 0$, all the eigenvalues of the Jacobian matrix constructed at $E_3(0,1,0)$ have negative real parts, indicating a

**Table 3. Stability Analysis of Evolutionary Stable Strategy.**

| Evolutionary Stable Strategy | Eigenvalues of the Jacobian Matrix | Stability |
|---|---|---|
| $E_1(0,0,0)$ | $\lambda_1 = -G_{cs}$ | (-, +, +) |
| | $\lambda_2 = G_{ei} + \varepsilon R_f$ | |
| | $\lambda_3 = Q_2 S_{ph} - Q_2' S_{ph} + Q_1 S_{pr} - Q_1' S_{pr}$ | |
| $E_2(0,0,1)$ | $\lambda_1 = -G_{cs} - G_{es} + \delta G_{es} + \delta G_f + G_t$ | (-, ×, -) |
| | $\lambda_2 = G_{ei} - R_{ec} + \varepsilon R_f + R_{rp}'$ | |
| | $\lambda_3 = -Q_2 S_{ph} + Q_2' S_{ph} - Q_1 S_{pr} + Q_1' S_{pr}$ | |
| $E_3(0,1,0)$ | $\lambda_1 = -G_{cs}$ | (-, -, +) |
| | $\lambda_2 = -G_{ei} - \varepsilon R_f$ | |
| | $\lambda_3 = G_{es} - \delta G_{es} - \delta G_f + Q_2 S_{ph} - Q_2' S_{ph} + Q_1 S_{pr} - Q_1' S_{pr}$ | |
| $E_4(1,0,0)$ | $\lambda_1 = G_{cs}$ | (+, +, +) |
| | $\lambda_2 = G_{ei} + \varepsilon R_f$ | |
| | $\lambda_3 = Q_2 S_{ph} - Q_2' S_{ph} + Q_1 S_{pr} - Q_1' S_{pr}$ | |
| $E_5(0,1,1)$ | $\lambda_1 = -G_{cs} - G_{es} + \delta G_{es} + \delta G_f$ | (-, ×, -) |
| | $\lambda_2 = -G_{ei} + R_{rc}' - R_{rp}' - \varepsilon R_f$ | |
| | $\lambda_3 = -G_{es} + \delta G_{es} + \delta G_f - Q_2 S_{ph} + Q_2' S_{ph} - Q_1 S_{pr} + Q_1' S_{pr}$ | |
| $E_6(1,0,1)$ | $\lambda_1 = G_{cs} + G_{es} - \delta G_{es} - \delta G_f - G_t$ | (×, +, -) |
| | $\lambda_2 = G_{ei} + \varepsilon R_f$ | |
| | $\lambda_3 = -Q_2 S_{ph} + Q_2' S_{ph} - Q_1 S_{pr} + Q_1' S_{pr}$ | |
| $E_7(1,1,0)$ | $\lambda_1 = G_{cs}$ | (+, -, +) |
| | $\lambda_2 = -G_{ei} - \varepsilon R_f$ | |
| | $\lambda_3 = (1-\delta) G_{es} - \delta G_f + S_e + Q_2 S_{ph} - Q_2' S_{ph} + Q_1 S_{pr} - Q_1' S_{pr}$ | |
| $E_8(1,1,1)$ | $\lambda_1 = G_{cs} + G_{es} - \delta G_{es} - \delta G_f$ | (×, -, -) |
| | $\lambda_2 = -G_{ei} - \varepsilon R_f$ | |
| | $\lambda_3 = -G_{es} + \delta G_{es} + \delta G_f - S_e - Q_2 S_{ph} + Q_2' S_{ph} - Q_1 S_{pr} + Q_1' S_{pr}$ | |

Note: "+" indicates a positive eigenvalue, "-" indicates a negative eigenvalue, and "×" indicates that the sign of the eigenvalue depends on the parameter values assigned.

stable equilibrium point. The corresponding evolutionary game strategy is $S_2 = \{$The local government does not provide subsidies to shippers, railway transport enterprises actively cooperate, and shippers actively choose road transport$\}$.

**Analysis:** The active cooperation of railway transport enterprises indicates that they have taken proactive measures to increase their competitiveness in the market, striving to attract shippers to choose rail transport. However, shippers, on the basis of market conditions and transportation costs, still tend to prefer road transport, likely because of its flexibility, convenience, and better adaptability to their freight needs.

This equilibrium point indicates that under the current conditions, the strategic combination of all parties has reached a relatively stable state. Even though railway enterprises take proactive measures, the lack of subsidies from the local government and shippers' preferences for road transport result in the overall game system gravitating toward this stable state.

**Scenario 3.** When $-G_{ei} + R'_{rc} - R'_{rp} - \varepsilon R_f < 0$, all the eigenvalues of the Jacobian matrix constructed at $E_5(0,1,1)$ have negative real parts, indicating a stable equilibrium point. The corresponding evolutionary game strategy is $S_3 = \{$The local government does not provide subsidies to shippers, railway transport enterprises actively cooperate, and shippers actively choose rail transport$\}$.

**Analysis:** The formation of this stable state can be understood as the market forces in the multiparty game reaching a new balance. The local government chooses not to provide subsidies, and thus, railway transport enterprises cannot rely on direct economic support from the government. Instead, these enterprises must attract shippers by enhancing their own market competitiveness. This improvement in competitiveness may manifest in various aspects, such as reducing freight rates, increasing transport efficiency, improving customer service, and expanding the service network.

In contrast, shippers, after weighing various factors under current market conditions, decide to choose rail transport. This decision may be influenced by the benefits brought about by the active cooperation of railway transport enterprises, such as lower transport costs, faster delivery, and greater cargo security. Additionally, in some scenarios, rail transport has unique advantages over road transport, and the choice of rail by shippers can be driven by considerations of their long-term interests.

Despite the absence of direct government subsidies, railway transport enterprises are still able to attract shippers through proactive market behavior. This active cooperation not only helps enterprises gain a larger market share but also strengthens their position in the market. In such a market environment, shippers, owing to their needs and cost considerations, choose rail transport as their primary transport mode, thereby bringing the entire system to a relatively stable equilibrium state. This equilibrium point reflects that even without government intervention, the market can achieve an effective balance through the self-adjustment of all parties involved.

**Scenario 4.** When $G_{cs} + G_{es} - \delta G_{es} - \delta G_f < 0$, all the eigenvalues of the Jacobian matrix constructed at $E_8(1,1,1)$ have negative real parts, indicating a stable equilibrium point. The corresponding evolutionary game strategy is $S_4 = \{$The local government provides subsidies to shippers, railway transport enterprises actively cooperate, and shippers actively choose rail transport$\}$.

**Analysis:** The local government reduces the cost of choosing rail transport for shippers through subsidies, incentivizing shippers to opt more for rail transport. This policy intervention effectively shifts shippers' transport preferences, encouraging more freight to be moved by rail. Railway transport enterprises actively cooperate with government policies by adopting measures to increase their competitiveness, such as improving service quality and efficiency, further attracting shippers to choose rail transport. Under the dual effect of subsidies and proactive measures by railway enterprises, shippers find that the overall cost of choosing

rail transport decreases, while they also receive higher-quality service, thus becoming more inclined to select rail transport.

This equilibrium point represents a cooperative relationship between railway transport enterprises and shippers under the incentive of government subsidies, forming a stable transport choice model. A positive feedback mechanism emerges among the local government, railway transport enterprises, and shippers, which reinforces itself and helps maintain the stability of this strategy combination.

## Simulation analysis

To verify the stability of the tripartite evolutionary game system, parameter values are assigned on the basis of the literature and relevant data, and numerical simulations of the model are conducted using Python 3.8.

### Parameter assignment

In this section, the parameters of the numerical simulation model are assigned values. In this study, all monetary values are expressed in Chinese yuan (RMB).

The following parameter values are based on references or other published sources:

$S_{pr} = 0.07$, $S_{ph} = 0.13$ [32]; $R_f = 2000$ [33]; $\delta = 45\%$ [20]; $Q_1 = 10\%$, $Q_2 = 90\%$ [34]; and $Q_1^{'} = 30\%$, $Q_2^{'} = 70\%$; according to the "Compilation of National Railway Statistical Data for 2023", $\varepsilon = 4\%$.

The following parameter assignments are based on operational data from a specific railway station segment in China:

$R_{rp}^{'} = 64$, $R_{ec}^{'} = 16$, $S_e = 100$, $G_{es} = 2000$, $G_f = 20000$, $G_{ei} = 100$, and $G_t = 30$.

The model is then analyzed under the above conditions.

### Initial evolutionary path

To verify the stability of the tripartite evolutionary game, numerical simulations of the model are conducted using Python 3.8, with the above parameter assignments. The simulation step size is set to 0.2, and the initial value is set to 0.1. In evolutionary game models, time variable $t$ represents a relative temporal scale, capturing the dynamic evolution of strategies rather than corresponding to actual physical time. Its unit is typically expressed as evolutionary steps or iteration counts, reflecting adjustments in strategy across different stages of system progression. Additionally, simulations conducted at varying time scales reveal distinct evolutionary dynamics; in the short term, the system may exhibit fluctuations, whereas in the long term, it generally converges toward a stable equilibrium. The simulation results are shown in Fig 5.

The simulation results indicate that under the condition that all the eigenvalues of the constructed Jacobian matrix have negative real parts, the equilibrium point for this dataset is (0,1,1). At this point, the following stable strategy set for the system exists: {the local government does not provide subsidies to shippers, railway transport enterprises actively cooperate, shippers actively choose rail transport}, which is consistent with the above stability analysis results of the equilibrium point.

### Initial value evolution simulation analysis

Under different initial values for x, y, and z, all subplots (a) to (e) in Fig 6 converge to the same stable state, with the corresponding evolutionary game strategy being $S_3 = $ {the local government does not provide subsidies to shippers, railway transport enterprises actively cooperate, shippers actively choose rail transport}, which is consistent with the above stability analysis of the equilibrium point.

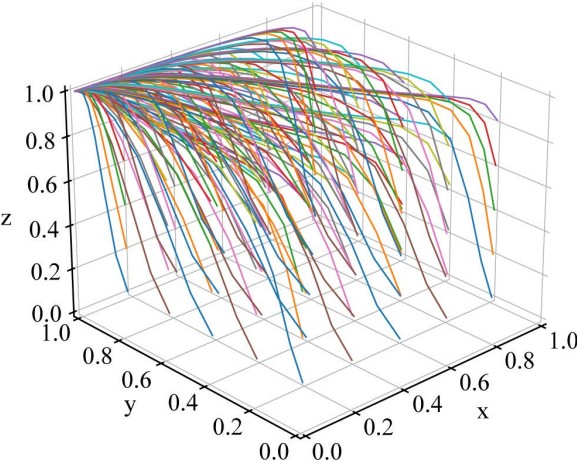

**Fig 5. Initial evolutionary path diagram.**

In subplot (a), when the initial value is set as (0.2,0.2,0.2), government intervention has a certain influence in the early stages. However, this influence gradually diminishes over time, and shippers' strategy shifts toward rail transport, while the market share of railway transport enterprises grows steadily. Similarly, in subplots (b) to (f), although the initial conditions vary and the evolutionary paths of the system differ slightly, the final equilibrium state remains fundamentally unchanged. Shippers still prefer a specific mode of transport, while the influence of government intervention gradually weakens and even disappears.

Fig 6 shows that the decision-making behavior of shippers evolves the fastest in this system, reaching a steady state first. Regardless of the initial conditions, shippers' decisions change rapidly in the early stages of evolution and stabilize within a short time. This finding indicates that shippers are strongly influenced by market conditions, costs, and transport efficiency when choosing a mode of transport and are able to make decisions quickly to adapt to changing environments.

In contrast to the rapid decision-making of shippers, the behavior of railway transport enterprises has evolved more slowly. Under different initial conditions, the market share of railway transport enterprises gradually increases, reaching stability over a relatively longer period. This finding suggests that the measures taken by railway transport enterprises in response to market competition take longer to have an impact than do those taken by other actors. Although the market share of railway transport enterprises eventually stabilizes and becomes dominant, the early adjustment process is relatively slow, indicating that these enterprises need more time to adjust strategies, optimize services, or enhance competitiveness in the face of market competition.

Initially, the local government may guide the market through policy intervention, allowing the market to achieve a self-regulating stable state under certain policy and competitive conditions. In the long run, enterprises and shippers in the market adjust according to their own interests and market conditions, leading the system toward stability. During this process, the role of the local government gradually diminishes, and the market mechanism becomes the primary regulating force.

## Sensitivity analysis of the simulation parameters

To eliminate the impact of the initial values of the selected parameters on the evolutionary game results of the system, target parameters are selected for corresponding sensitivity analysis.

**Sensitivity simulation analysis of the road to rail freight volume ratio parameters $Q_1$ and $Q_2$.** In September 2018, the General Office of the State Council of the People's Republic

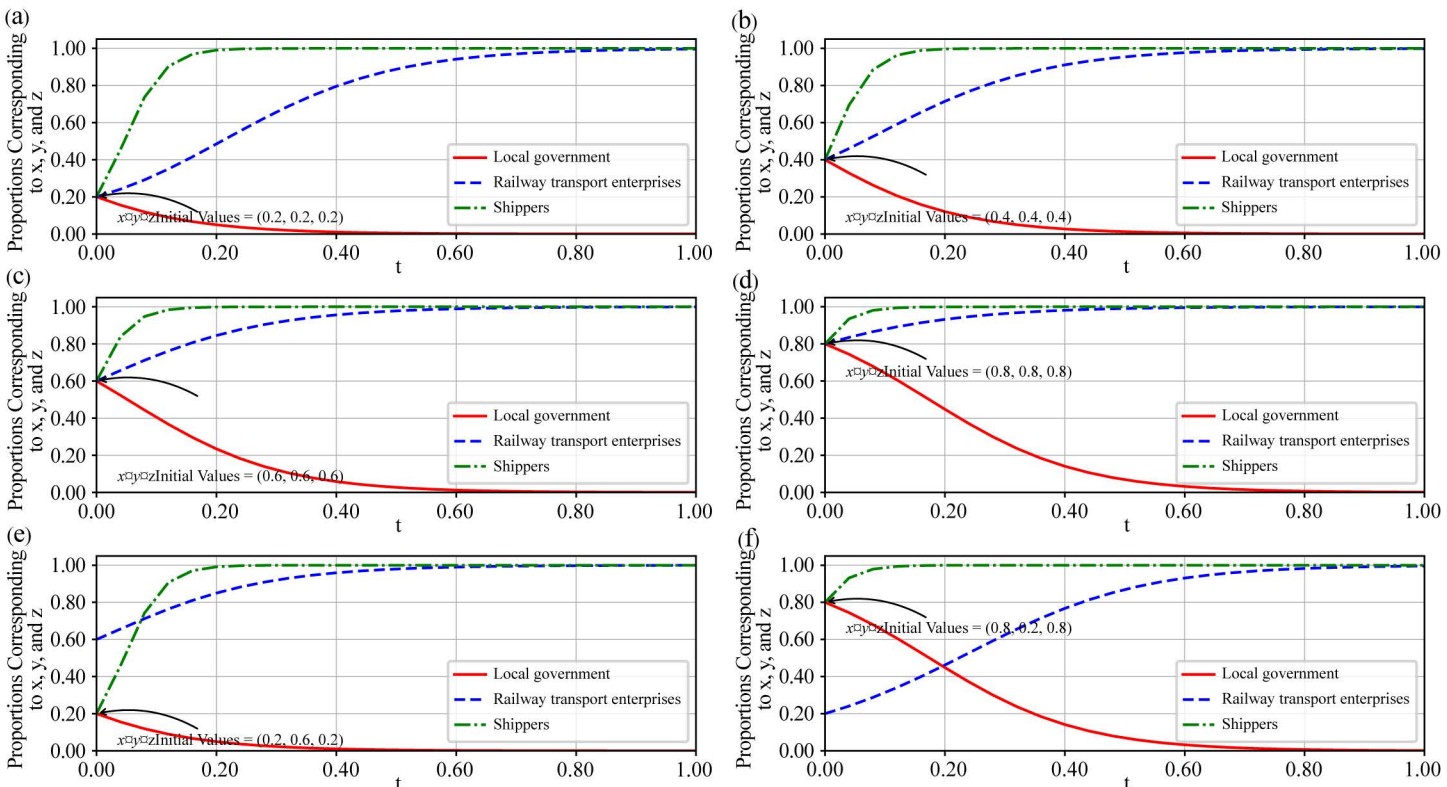

**Fig 6. Evolutionary simulation graph of different initial values for x, y, and z.**

of China officially issued the Three Year Action Plan for Promoting Transport Structure Adjustment (2018–2020), which aimed to increase the national railway freight volume by 1.1 billion tons (30% growth) compared with 2017 via a three-year concentrated campaign.

To investigate the impact of road freight volume $Q_1$ and rail freight volume $Q_2$ on the evolutionary game results of the system, the parameters are set to different values for the road to rail freight volume ratio, while the other parameters are kept constant as follows: $Q_1 = 10\%$, $Q_2 = 90\%$; $Q_1 = 20\%$, $Q_2 = 80\%$; $Q_1 = 30\%$, $Q_2 = 70\%$; $Q_1 = 40\%$, $Q_2 = 60\%$; and $Q_1 = 50\%$, $Q_2 = 50\%$. The simulation results are shown in Fig 7.

As shown in Fig 7, under different initial values of $Q_1$ and $Q_2$, the simulation ultimately converges to the same stable state, specifically at point (0,1,1), which is consistent with the above stability analysis results of the equilibrium point.

Currently, the ratio of rail freight volume to road freight volume in China is approximately 1:9, respectively, and the time to convergence to a stable state in the evolutionary game is relatively slow. When the proportion of rail freight gradually increases, the system converges to a stable state more quickly. This finding indicates that in the current transport system, promoting the shift of freight transport from road to rail is necessary to enhance the overall efficiency of the system. Increasing the proportion of rail freight not only reduces overall transportation costs but also decreases carbon emissions and other environmental pollution, as well as reducing the number of accidents and risks associated with road transport.

**Sensitivity simulation analysis of government subsidy parameter $G_{es}$ for shippers.** To investigate the impact of government subsidies $G_{es}$ for shippers on the evolutionary game

results of the system, while the other parameters are kept constant, subsidy parameter $G_{es}$ is assigned values of $G_{es} = 2000$, $G_{es} = 3000$, $G_{es} = 5000$, $G_{es} = 8000$, and $G_{es} = 10000$. The simulation results are shown in Fig 8.

As shown in Fig 8, under different initial values of $G_{es}$, the simulation ultimately converges to the same stable state, specifically at point (0,1,1), which is consistent with the above stability analysis results of the equilibrium point.

The growth trend and turning points of the curve can reflect the stability and sensitivity of the system under different conditions. The curve corresponding to a lower subsidy value, $G_{es}$, tends to be lower in the range of changes for y and z, indicating that the system takes longer to reach an equilibrium state under weaker stimuli. As the subsidy value $G_{es}$ appropriately increases, the corresponding y and z values of the curve increase, suggesting that the system reaches equilibrium more quickly. However, as $G_{es}$ continues to increase, the speed at which the system reaches equilibrium begins to decrease, indicating that the marginal utility of higher subsidies diminishes compared with that of lower subsidies. In the early stages of market development, government subsidies are needed to influence the strategic behavior of shippers. However, as the system continues to evolve, the effectiveness of subsidies gradually decreases until, eventually, the local government no longer provides subsidies to shippers, and the tripartite evolutionary game system reaches an equilibrium state.

**Sensitivity simulation analysis of probability ´ of the local government supervising shippers receiving subsidies.** To investigate the impact of probability $\delta$ of the local government supervising shippers receiving subsidies on the evolutionary game results of the system, the parameters are set to different values for probability $\delta$ of the local government supervising shippers receiving subsidies, while the other parameters are kept constant: $\delta = 15\%$, $\delta = 30\%$, $\delta = 45\%$, $\delta = 50\%$, and $\delta = 55\%$. The simulation results are shown in Fig 9.

As shown in Fig 9, when the probability $\delta$ of the local government supervising shippers receiving subsidies is less than approximately 50%, the system converges to the point (0,1,1), which is consistent with the above stability analysis results of the equilibrium point. However, when the probability $\delta$ exceeds approximately 50%, the system fails to converge normally.

The simulation results demonstrate that the probability of local government supervision $\delta$ significantly influences the evolutionary outcomes of the system. When the supervision probability $\delta < 50\%$, shippers exhibit a greater tendency to accept subsidies, enabling the system to converge smoothly to the stable equilibrium point (0,1,1). Furthermore, a comparative analysis of different values of $\delta$ reveals that as the supervision probability approaches

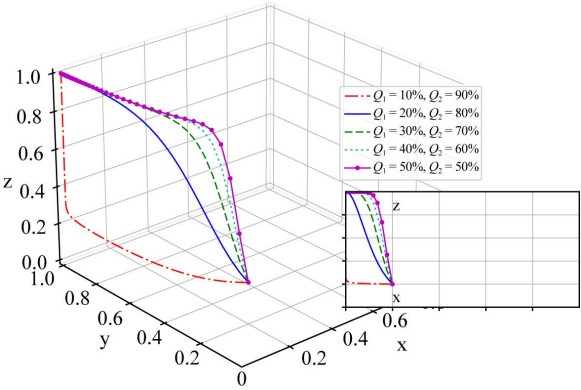

**Fig 7. Sensitivity Analysis of the Parameters for Road to rail Freight Volume Ratio $Q_1$ and $Q_2$.**

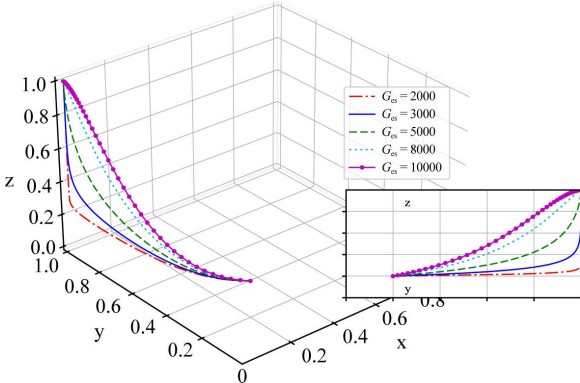

**Fig 8. Sensitivity Analysis of Government Subsidy Parameter $G_{es}$ for Shippers.**

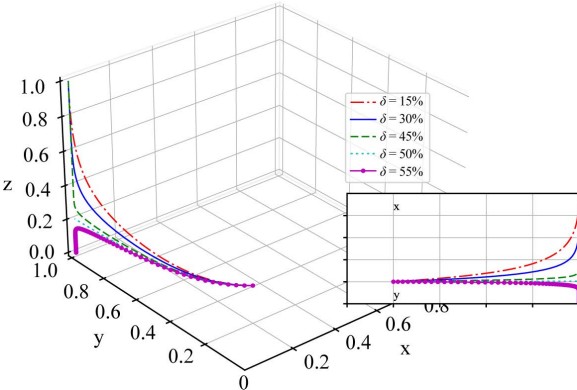

**Fig 9. Sensitivity Analysis of Probability ´ of the Local Government Supervising Shippers Receiving Subsidies.**

50%, the system converges to equilibrium at an accelerated rate. This finding suggests that a moderate increase in supervision intensity can facilitate the rapid convergence of the system to its equilibrium state, effectively constraining the behavior of shippers while reducing the associated regulatory costs for the government. However, when $\delta$ exceeds 50%, the system fails to achieve stable equilibrium and instead exhibits instability or oscillatory behavior. This finding indicates that excessive supervision intensity may discourage shippers from accepting subsidies, thereby disrupting the established cooperative framework and undermining the dynamic stability of the system. Consequently, local governments must strike a balance when formulating regulatory strategies, ensuring that supervision intensity is sufficiently robust to promote convergence and stability while avoiding overregulation, which could jeopardize system performance and equilibrium.

**Sensitivity simulation analysis of penalty $G_f$ imposed on shippers for noncompliant behavior in receiving subsidies if they fail to carry out the related freight transport as required.** To investigate the impact of penalty $G_f$ imposed on shippers for noncompliant behavior in receiving subsidies if they fail to carry out the related freight transport as required on the evolutionary game results of the system, parameter $G_f$ is set to different values, while the other parameters are kept constant— $G_f = 2000$ , $G_f = 5000$ , $G_f = 10000$ , $G_f = 20000$ , and $G_f = 30000$ . The simulation results are shown in Fig 10.

The simulation results indicate that when penalty $G_f$ imposed on shippers for noncompliant behavior in receiving subsidies is less than approximately 20,000, the system consistently converges to a stable equilibrium state at the point (0,1,1), aligning with the equilibrium stability analysis discussed earlier. However, when $G_f$ exceeds 20,000, the system begins to deviate from its convergence trajectory and fails to achieve equilibrium.

This outcome suggests that excessively high penalties may trigger resistance or behavioral shifts among shippers, thereby disrupting the system's inherent dynamic equilibrium and potentially leading to system instability or failure. Notably, within the penalty range of 2,000–10,000, variations in the penalty intensity have a minimal effect on the convergence speed of the system. In contrast, when the penalty approaches 20,000, the system's convergence speed increases significantly. Therefore, policymakers must carefully balance penalty intensity and system stability, setting an appropriate upper threshold for $G_f$ to effectively regulate shippers' behavior while ensuring system stability and operational efficiency.

## Inspiration and discussion

### Alignment with and divergence from prior studies

This study investigates the dynamic interactions present among the local government, railway transport enterprises, and shippers under the road to rail strategy in China through an evolutionary game model. The findings reveal both alignments and divergences with existing research, offering insights into the strategic adjustments of each stakeholder and the temporal evolution of stakeholder interactions. The following aims to explain the consistencies and differences between the current study and existing literature.

**Heterogeneity aspect.** This study selects local governments, shippers, and railway transport enterprises as the participants in the evolutionary game. In contrast to previous road to rail studies—such as Yang and Zhang [21], who chose to examine local governments, road transport enterprises, and railway transport enterprises, and Zhang and Xu [23], who chose to consider local governments, intermediary platforms, and the market—this research incorporates shippers as direct participants. As the consumers of transportation services, shippers' decisions to opt for rail or road transport directly determine market demand and the allocation of transportation resources. Thus, including shippers in the game framework yields a more realistic depiction of the supply–demand dynamics and decision-making

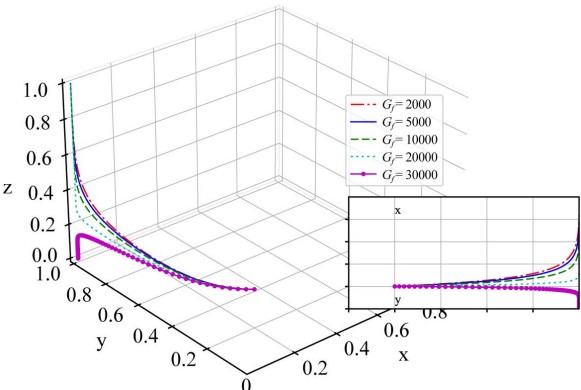

**Fig 10. Sensitivity Analysis of Penalty $G_f$ Imposed on Shippers for Noncompliant Behavior in Receiving Subsidies if they Fail to Carry Out the Related Freight Transport as Required.**

processes that are inherent in the transportation market. In practice, the success of railway transport enterprises in attracting shippers and increasing market share largely depends on shippers' acceptance of railway services. Thus, integrating shippers into the evolutionary game facilitates the analysis of how their choice behaviors—under varying policies and market conditions—affect the overall transformation of transportation modes, thereby providing more targeted theoretical support for policy formulation and corporate strategy. Moreover, by employing a game-theoretical model combined with numerical simulations, the current study not only dynamically captures the interactive evolution among multiple stakeholders but also quantitatively elucidates the influence of various parameters on the final equilibrium. This methodological approach effectively addresses the limitations of traditional policy research on Chinese railways (e.g., Lin [10] and Wang et al. [8]) with respect to dynamism, realism, and quantitative analysis.

**Consistency aspect.** The theme of "the transition from road to rail" adopted in this study reflects the urgent need for the transformation of China's transportation structure and green development. Research by Rotaris et al. [12] and Wisetjindawat et al. [13] has further indicated that this theme has garnered global attention. With an increasing emphasis being placed on energy conservation, emission reduction, green transportation, and sustainable development in China, railway transport—as an efficient and environmentally friendly mode—is gradually receiving enhanced policy support and market favor. In recent years, the Chinese government has vigorously promoted high-speed rail and modern railway network construction, thereby significantly improving the service capacity and geographic coverage of railway transport. Coupled with national strategies such as the Belt and Road Initiative, the development of a rail-dominated comprehensive transportation system not only alleviates road congestion and reduces carbon emissions but also fosters coordinated regional logistics and economic transformation. One of the conclusions of the current study is that government subsidies play a crucial role during the initial phase by reducing the financial burden on shippers and incentivizing the adoption of railway transport. However, the marginal benefits of these subsidies diminish over time, gradually transferring regulatory responsibilities to market mechanisms. Feng et al. [20] similarly concluded—within the context of the evolution of the China Railway Express market—that only by implementing measures against fraudulent subsidy claims and ensuring consistent subsidy withdrawal between platform enterprises and shippers can the market eventually evolve to a subsidy-free state. Additional studies by Lin [10] and Feng [11] have also demonstrated that proactive government actions can effectively drive adjustments in transportation structure. By designing appropriate subsidy exit mechanisms and enhancing regulatory oversight, the government can guide the market to progressively assume control, thereby achieving a smooth transition from government dominance to market self-regulation. This process not only ensures the optimal upgrading of the transportation structure but also substantiates the pivotal role of early government intervention in steering subsequent market-oriented development.

## Theoretical contributions

This study makes significant contributions to the theoretical understanding of the road to rail strategy in China through the application of evolutionary game modeling. First, this study introduces a dynamic analytical framework that captures the adaptive strategies of local governments, railway transport enterprises, and shippers over time. This approach represents a substantial advancement over conventional static models, enabling a more comprehensive understanding of stakeholders' strategic evolution under varying policy and market conditions. Second, this study offers insights into the optimal timing for subsidy withdrawal, emphasizing the need to align policy phases with market maturity and self-regulation

capacity. By incorporating temporal dynamics, the findings address a critical research gap, where previous studies often overlook the time-sensitive nature of subsidy effectiveness. Third, this study highlights the asymmetry in strategy adjustment speeds among stakeholders. Shippers respond rapidly to cost incentives and policy changes, whereas railway transport enterprises require longer periods to optimize services and adapt operational strategies. This finding underscores the importance of synchronized policy interventions and market mechanisms to ensure balanced progress across stakeholders.

Collectively, these contributions deepen the theoretical foundation for understanding the dynamic interactions underpinning transport structure adjustments, providing a robust analytical basis for future research.

## Practical recommendations

The findings of this study offer several practical recommendations for effectively promoting the shift from road to rail in freight transportation.

For local governments, focusing on early-stage subsidies to accelerate behavioral shifts among shippers is recommended. These subsidies should be targeted, time bound, and performance driven for maximum effectiveness. As market mechanisms gradually strengthen, subsidies should be phased out systematically to prevent long-term dependence. Furthermore, periodic reviews of subsidy performance and robust supervision mechanisms are essential to ensure transparency, efficiency, and accountability.

For railway transport enterprises, the primary focus should be on service quality enhancement, which includes improving operational efficiency, delivery reliability, and customer responsiveness. Enterprises must prioritize investments in digital infrastructure and adopt flexible logistics solutions to optimize freight operations. Building long-term partnerships with shippers through performance-based incentives and tailored logistics services can further strengthen market stability and foster mutual trust for enterprises.

For shippers, recognizing the long-term economic and environmental benefits of rail transport is essential. Shippers should actively leverage government incentives and collaborate with railway enterprises to optimize logistics planning and cost structures. Effective engagement with government policy frameworks can enable shippers to maximize the advantages offered by subsidy programs and improved rail services.

Furthermore, collaborative platforms involving the local government, railway enterprises, and shippers should be established to facilitate transparent communication, efficient problem-solving, and strategic alignment. Such platforms can enhance policy implementation and allow for operational bottlenecks to be addressed more effectively.

## Conclusions

In this study, a tripartite evolutionary game system involving the local government, railway transport enterprises, and shippers is constructed. On the basis of real data collection combined with theoretical analysis and simulation models, the stability of decision-making behaviors among the three parties is analyzed, the conclusions of which are presented below.

(1) **Shippers adjust decisions rapidly on the basis of market changes**: The strategy evolution speed of shippers is significantly greater than that of the local government and railway transport enterprises, reflecting shippers' high-level sensitivity and adaptability to market changes. This rapid adjustment is not only driven by shippers' direct interest in optimizing transportation efficiency and cost but also indicates their flexible response to policy changes. This quick response may have a dual impact on market competition; on the one hand, it can push railway transport enterprises to continuously optimize service quality

and transportation prices to maintain an advantageous position in the game, and on the other hand, it encourages local governments to adjust policy support in a timely manner, forming a dynamic evolution mechanism of mutual incentives among policies, enterprises, and the market, thus achieving the optimal goal of transport structure adjustment.

(2) **There is a low current rail freight volume but a noticeable trend in absorbing road freight**: Although the current rail freight volume accounts for only approximately 10% of the total, with road transport accounting for 90%, the simulation results reveal an increasingly significant trend of freight shifting from road to rail under local government policy incentives. When the rail freight proportion increases to approximately 30%, the system reaches equilibrium more quickly. This shift not only effectively reduces overall transportation costs but also significantly decreases negative externalities such as carbon emissions and traffic accidents during transportation, providing notable social benefits in terms of environmental protection and public safety. Moreover, as the amount of rail freight increases, economies of scale begin to emerge, further consolidating the competitive advantage of rail transport. This finding indicates that through scientific policy guidance, the road to rail shift can reduce overall societal costs while optimizing the transport structure, thus creating a synergy of economic, environmental, and social benefits.

(3) **An integrated strategy that combines government subsidies, supervision intensity, and penalty mechanisms is crucial for effectively promoting and sustaining the modal shift from road to rail transport**: In the initial evolutionary phase, government subsidies serve as a fundamental incentive, significantly accelerating system convergence and fostering trust in rail-based logistics systems. The simulation results reveal that increasing subsidy levels enhance shippers' willingness to adopt rail transport; however, as market maturity advances and trust between stakeholders solidifies, the marginal benefits of subsidies diminish. A prolonged reliance on subsidies may erode market self-regulation, necessitating a phased withdrawal strategy to balance fiscal responsibility and long-term sustainability. Additionally, the supervision probability ($\delta$) of local governments plays a pivotal role in stabilizing system dynamics. Moderate supervision accelerates convergence and promotes shipper compliance, whereas excessive supervision intensity disrupts equilibrium, leading to system instability. Likewise, penalties ($G_f$) for noncompliant behavior influence convergence outcomes, with moderate penalties facilitating stability and excessively high penalties triggering resistance and destabilization. Thus, achieving an optimal balance among these three mechanisms is essential. Policymakers should adopt a dynamic approach, offering well-targeted subsidies in the initial phase, implementing moderate supervision intensity, and carefully calibrating penalty thresholds to maintain system stability. This integrated strategy ensures smooth convergence, enhances stakeholder cooperation, and fosters a resilient, self-regulating transportation system aligned with the long-term objectives of the road to rail initiative.

While this study offers valuable insights, several avenues for future research remain open for exploration.

First, future studies could consider incorporating additional stakeholders, such as logistics service providers, financial institutions, and environmental agencies, into the evolutionary game model. This expanded perspective would provide a more holistic understanding of the multistakeholder dynamics driving the modal shift.

Second, further research could focus on developing refined subsidy optimization models to predict the long-term impacts of different subsidy strategies. Such models could assist

policymakers in designing time-sensitive intervention strategies tailored to regional and market-
specific conditions.

Finally, multimodal transport integration remains a key area for investigation. Understanding how road and rail transport systems can complement one another within an integrated logistics network is essential for addressing current bottlenecks and achieving seamless transport operations.

## Supporting information

**S1 File. Supporting information.**
(DOCX)

## Author contributions

**Conceptualization:** Shuai Liu.

**Data curation:** Shuai Liu.

**Formal analysis:** Shuai Liu.

**Funding acquisition:** Guangzhi Jia.

**Methodology:** Shuai Liu, Guangzhi Jia.

**Writing – original draft:** Shuai Liu.

**Writing – review & editing:** Guangzhi Jia.

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
