## [Decision Letter · Decision Letter 0]

11 Dec 2024

PONE-D-24-54352Promoting the Modal Shift of Freight from Road to Rail in China An Evolutionary Game and Simulation StudyPLOS ONE

Dear Dr. Liu,

Thank you for submitting your manuscript to PLOS ONE. After careful consideration, we feel that it has merit but does not fully meet PLOS ONE’s publication criteria as it currently stands. Therefore, we invite you to submit a revised version of the manuscript that addresses the points raised during the review process.

We look forward to receiving your revised manuscript.

Kind regards,

Mohamed Rafik N. Qureshi, Ph.D.

Academic Editor

PLOS ONE

Journal Requirements:

When submitting your revision, we need you to address these additional requirements. 1. Please ensure that your manuscript meets PLOS ONE's style requirements, including those for file naming. The PLOS ONE style templates can be found at https://journals.plos.org/plosone/s/file?id=wjVg/PLOSOne_formatting_sample_main_body.pdf and https://journals.plos.org/plosone/s/file?id=ba62/PLOSOne_formatting_sample_title_authors_affiliations.pdf 2. Please note that PLOS ONE has specific guidelines on code sharing for submissions in which author-generated code underpins the findings in the manuscript. In these cases, we expect all author-generated code to be made available without restrictions upon publication of the work. Please review our guidelines at https://journals.plos.org/plosone/s/materials-and-software-sharing#loc-sharing-code and ensure that your code is shared in a way that follows best practice and facilitates reproducibility and reuse. 3. We suggest you thoroughly copyedit your manuscript for language usage, spelling, and grammar. If you do not know anyone who can help you do this, you may wish to consider employing a professional scientific editing service.  The American Journal Experts (AJE) (https://www.aje.com/) is one such service that has extensive experience helping authors meet PLOS guidelines and can provide language editing, translation, manuscript formatting, and figure formatting to ensure your manuscript meets our submission guidelines. Please note that having the manuscript copyedited by AJE or any other editing services does not guarantee selection for peer review or acceptance for publication.  Upon resubmission, please provide the following: The name of the colleague or the details of the professional service that edited your manuscript A copy of your manuscript showing your changes by either highlighting them or using track changes (uploaded as a *supporting information* file) A clean copy of the edited manuscript (uploaded as the new *manuscript* file)”. 4. We note that your Data Availability Statement is currently as follows: [All relevant data are within the manuscript and its Supporting Information files.] Please confirm at this time whether or not your submission contains all raw data required to replicate the results of your study. Authors must share the “minimal data set” for their submission. PLOS defines the minimal data set to consist of the data required to replicate all study findings reported in the article, as well as related metadata and methods (https://journals.plos.org/plosone/s/data-availability#loc-minimal-data-set-definition). For example, authors should submit the following data: - The values behind the means, standard deviations and other measures reported;- The values used to build graphs;- The points extracted from images for analysis. Authors do not need to submit their entire data set if only a portion of the data was used in the reported study. If your submission does not contain these data, please either upload them as Supporting Information files or deposit them to a stable, public repository and provide us with the relevant URLs, DOIs, or accession numbers. For a list of recommended repositories, please see https://journals.plos.org/plosone/s/recommended-repositories. If there are ethical or legal restrictions on sharing a de-identified data set, please explain them in detail (e.g., data contain potentially sensitive information, data are owned by a third-party organization, etc.) and who has imposed them (e.g., an ethics committee). Please also provide contact information for a data access committee, ethics committee, or other institutional body to which data requests may be sent. If data are owned by a third party, please indicate how others may request data access.

Additional Editor Comments:

The manuscript entitled "Promoting the Modal Shift of Freight from Road to Rail in China An Evolutionary Game and Simulation Study" should be revised as per the reviewers comments for further action.

Reviewers' comments:

Reviewer's Responses to Questions

**Comments to the Author**

1. Is the manuscript technically sound, and do the data support the conclusions?

Reviewer #1: Partly

Reviewer #2: Yes

Reviewer #3: Yes

2. Has the statistical analysis been performed appropriately and rigorously? 

Reviewer #1: N/A

Reviewer #2: Yes

Reviewer #3: Yes

3. Have the authors made all data underlying the findings in their manuscript fully available?

Reviewer #1: No

Reviewer #2: Yes

Reviewer #3: Yes

4. Is the manuscript presented in an intelligible fashion and written in standard English?

Reviewer #1: Yes

Reviewer #2: No

Reviewer #3: Yes

5. Review Comments to the Author

Reviewer #1: C1) I recommend that authors number their equations. This makes referencing equations easier both in the text and for reviews.

C2) In subchapter 3.1, in the third and fourth equations, which give F(x) and dF(x)/dx, it is recommended to repeat the expected payoff expressions Eτ1 and Eτ2. Only the final forms for F(x) and dF(x)/dt are recommended. The observations are also valid for the corresponding equations in chapters 3.2 and 3.3.

C3) In subchapter 3.1, please check the sign before the parenthesis in the third equation, which gives F(x) in the first square bracket.

C4) At the beginning of chapter 3, we recommend specifying one or more well-known references in games and simulations of the type described below, which show the working algorithm, so that the reader can refer to them for terms of reference and developments.

C5) In Figure 2, please explain VA1, VA2. Similar for VB1, VB2, VC1, VC2, from Figures 3 and 4.

C6) It is good for payments and costs to have specified units of measurement: for example: Rf= 2000? , yuan, dollars, euros? Pag. 19). Also Ges =2000 ?, Gf= 20000 ? If they are dimensionless, specify!

A clarification on this aspect only appears in conclusion 3. It is late. You make the statement at the beginning of the mathematical model and it is clear for the rest of the article.

C7) For the Figure 6: Please explain what the variable on the abscissa, t, is. Is it the current time or a scaled time? What is the unit of measurement of the variable represented on the graphs in Figure 6? The time variable is mentioned in many places in the article. It would be useful, for a better understanding of the simulation, of the process, to explain which of the parameters that appear in the process depend on time, and, if possible, in what way (continuous, discontinuous, with jumps?).

Explaining the role of time in this game can also bring estimates of the period of reaching a state of stability, of the expectations of governments to be able to give up sponsorships.

C8) The conclusions cannot be fully proven, more precisely the statements regarding the speed of stabilization, the rapidity of strategy changes and the response of the participants to the game. Therefore, we suggest an example that includes strategy changes over time, continuously or in steps. Is there an optimal strategy over time, for example for the fastest stabilization?

C9) The sources of numerical data in chapter 4, subchapter 4.1, must be listed in the reference list and cited normally. This is the reason for my answer to question 3. The deficiency is easy to eliminate.

Observations C1)-C9) are the reasons for the answer to question 1 of the review form. Authors should address the issues and, as they see fit, make or not make changes.

The language can be improved (English and the academic language).

Reviewer #2: Dear Authors,

Greetings!

I am pleased to have reviewed the comprehensive study you have submitted to PLOS ONE. Your research is of high quality, but to further enhance the academic quality and publication prospects of the paper, I suggest making the following revisions.

1. Introduction: It is crucial to clearly state the following aspects in the introduction:

- The main research questions;

- The primary objectives of the study;

- The gaps in the literature that this study aims to address;

- The motivation behind the study;

- The innovation and novelty of the study compared to existing work.

2. Sensitivity Analysis of Parameters: In the section on sensitivity analysis of parameters, it is recommended to include analysis of the following more critical parameters:

- The probability of government supervision;

- The penalty for shippers who apply for subsidies but fail to carry out the related freight transport as required;

- The government's investment subsidies for railway companies

3. Article Structure: It is suggested to add some managerial implications after the conclusion to assist in the shift of cargo transportation from road to rail.

4. Grammar and Formatting: There are minor grammatical errors in the paper, and I recommend linguistic polishing to improve the readability of the manuscript.

5. Citation Formatting: There are citation formatting errors in Section 3.4 and Section 4.1. Please carefully proofread and correct these formatting issues.

After the above minor revisions, I will support the publication of this study. I look forward to the improvements in your manuscript and wish you greater success in your research endeavors.

Best regards!

Reviewer #3: The authors have selected a relevant and significant topic, presenting it in a clear manner while demonstrating a solid grasp of the fundamental concepts. They have effectively highlighted the paper's contribution. The technical rigor of the applied processes and statistical methods is satisfactory. My only concern is the discussion section, I recommend elaborating on your results by explaining how they align with or diverge from prior studies. Provide potential explanations for why your findings might support or contradict previous research and explore their implications. This will offer readers a more comprehensive understanding of the topic.

6. PLOS authors have the option to publish the peer review history of their article (what does this mean? ). If published, this will include your full peer review and any attached files.

**Do you want your identity to be public for this peer review?** For information about this choice, including consent withdrawal, please see our Privacy Policy .

Reviewer #1: **Yes: ** Petru Cardei

Reviewer #2: No

Reviewer #3: No

---

## [Author Response · Author response to Decision Letter 1]

19 Jan 2025

Dear Dr. Mohamed Rafik N. Qureshi,

We sincerely appreciate your constructive feedback and the valuable comments from the reviewers regarding our manuscript, "Promoting the Modal Shift of Freight from Road to Rail in China: An Evolutionary Game and Simulation Study" (PONE-D-24-54352).

Following your suggestions and the reviewers’ comments, we have carefully revised the manuscript to address all the points raised. Below is a summary of the changes made:

Rebuttal Letter

We have prepared a detailed rebuttal letter addressing each point raised by the reviewers. This document is attached under the file name "Response to Reviewers".

Revised Manuscript with Track Changes

We have included a marked-up version of the revised manuscript that highlights all the changes made in response to the reviewers' comments. This file is attached under the name "Revised Manuscript with Track Changes".

Clean Manuscript

A clean version of the revised manuscript without track changes is also attached under the file name "Manuscript".

Editing and Formatting

In accordance with the journal's requirements, we utilized the editing services of AJE (American Journal Experts). The manuscript was edited by Catherine Zettel Nalen, MS, a Research Communication Partner, who provided Premium Editing and Manuscript Formatting services. This ensured that the language, grammar, and formatting adhered to the standards of PLOS ONE.

Additional Data and Clarifications

We have ensured that all raw data necessary to replicate the study’s findings are included in the manuscript or its supporting information files.

The sources of numerical data used in Chapter 4, Subchapter 4.1, have been properly cited and included in the reference list, as suggested by Reviewer #1.

Clarifications regarding variables, units of measurement, and time parameters in the mathematical model have been added to enhance clarity and reproducibility.

Revisions Based on Reviewers’ Comments

We revised the introduction to clearly state the research questions, objectives, literature gaps, motivation, and novelty, as suggested by Reviewer #2.

Sensitivity analyses of additional critical parameters, such as government supervision probability and penalty mechanisms, have been included.

Managerial implications have been added to the conclusion section to provide actionable insights for promoting the shift from road to rail freight transport.

The discussion section has been expanded to align with Reviewer #3’s request, elaborating on how our findings compare to prior studies and their implications.

Figures and Equations

Figures and equations were renumbered and revised for consistency and clarity.

Explanations of variables and units in figures, such as Figure 6, were added to ensure proper understanding.

We hope that the revised manuscript meets the expectations of the journal and reviewers. Should you require any further information or clarification, please do not hesitate to contact us.

Kind regards,

ShuaiLiu

---

## [Decision Letter · Decision Letter 1]

27 Jan 2025

PONE-D-24-54352R1Promoting the Modal Shift of Freight from Road to Rail in China An Evolutionary Game and Simulation StudyPLOS ONE

Dear Dr. Liu,

Thank you for submitting your manuscript to PLOS ONE. After careful consideration, we feel that it has merit but does not fully meet PLOS ONE’s publication criteria as it currently stands. Therefore, we invite you to submit a revised version of the manuscript that addresses the points raised during the review process.

We look forward to receiving your revised manuscript.

Kind regards,

Mohamed Rafik N. Qureshi, Ph.D.

Academic Editor

PLOS ONE

Journal Requirements:

Additional Editor Comments:

The manuscript entitled 'Promoting the Modal Shift of Freight from Road to Rail in China An Evolutionary Game and Simulation Study' needs further revision as per the reviewer's comments.

Reviewers' comments:

Reviewer's Responses to Questions

**Comments to the Author**

1. If the authors have adequately addressed your comments raised in a previous round of review and you feel that this manuscript is now acceptable for publication, you may indicate that here to bypass the “Comments to the Author” section, enter your conflict of interest statement in the “Confidential to Editor” section, and submit your "Accept" recommendation.

Reviewer #1: All comments have been addressed

Reviewer #3: (No Response)

2. Is the manuscript technically sound, and do the data support the conclusions?

Reviewer #1: Yes

Reviewer #3: Yes

3. Has the statistical analysis been performed appropriately and rigorously? 

Reviewer #1: Yes

Reviewer #3: Yes

4. Have the authors made all data underlying the findings in their manuscript fully available?

Reviewer #1: Yes

Reviewer #3: Yes

5. Is the manuscript presented in an intelligible fashion and written in standard English?

Reviewer #1: Yes

Reviewer #3: Yes

6. Review Comments to the Author

Reviewer #1: I checked the authors' approach to all my observations. I found that the authors responded very kindly to all observations. There were also additions (of course from the other reviewers) in addition. The work is suitable for publication.

The development of the apparatus for improving transport by rail and road is a necessity in the world now. That is why there are many directions for further development and game theory has the most important role. However, the simulation will have to address the side of "dishonest" players at some point. However, the theme of the work is very generous and it would be good to transform it into software that would regulate the organization of transport in certain regions. However, it is necessary to avoid seizing transport management in a region or area, using exclusively a certain software. There must be alternatives.

Reviewer #3: The authors have selected a relevant and significant topic, presenting it clearly and demonstrating a solid grasp of the fundamental concepts. They have effectively highlighted the paper's contribution, and the technical rigor of the applied processes and statistical methods remains satisfactory. However, I noticed that the changes requested regarding the discussion section were not fully addressed. Specifically, I had recommended elaborating on your results by explaining how they align with or diverge from prior studies. Additionally, I suggested providing potential explanations for why your findings might support or contradict previous research and exploring their implications. Unfortunately, these aspects remain insufficiently developed.

I strongly encourage revisiting the discussion section to incorporate these elements. Expanding on how your findings connect to existing literature will not only enhance the robustness of your manuscript but also provide readers with a deeper and more nuanced understanding of the topic. Addressing this will greatly strengthen the overall contribution and impact of your work.

7. PLOS authors have the option to publish the peer review history of their article (what does this mean? ). If published, this will include your full peer review and any attached files.

**Do you want your identity to be public for this peer review?** For information about this choice, including consent withdrawal, please see our Privacy Policy .

Reviewer #1: **Yes: ** Petru Cardei

Reviewer #3: No

---

## [Author Response · Author response to Decision Letter 2]

12 Feb 2025

Dear Editor and Reviewers,

We thank the editor and the reviewers for their valuable time and effort in our paper, which has helped us to improve the quality of our paper. In the following, we address each of the reviewers' comments and suggestions. We have revised our manuscript accordingly and believe that the revised version addresses all the concerns raised by the reviewers. Changes made in the first revision have been highlighted in yellow, and changes made in the second revision have been highlighted in green to distinguish them. We hope that our revised manuscript meets the standards of your prestigious journal.

Reviewer #1:

Thank you for your positive feedback and valuable suggestions regarding our work. We will further explore and refine the relevant aspects in our future research.

Thank you again for your support and encouragement!

If you have any other questions or concerns, please feel free to contact the authors and we will respond and improve them seriously.

Reviewer #2:

The authors appreciate your statement on our work. We have revised the paper according to your comments. The yellow marked parts are the corresponding revisions.

1. Introduction: It is crucial to clearly state the following aspects in the introduction:

- The main research questions;

- The primary objectives of the study;

- The gaps in the literature that this study aims to address;

- The motivation behind the study;

- The innovation and novelty of the study compared to existing work.

Response:

We appreciate your careful review of our manuscript. We have added the relevant content in the "Introduction" section.

Action:

Please see page 7 and 8.

2. Sensitivity Analysis of Parameters: In the section on sensitivity analysis of parameters, it is recommended to include analysis of the following more critical parameters:

- The probability of government supervision;

- The penalty for shippers who apply for subsidies but fail to carry out the related freight transport as required;

- The government's investment subsidies for railway companies.

Response:

We are very pleased that you raised questions regarding the sensitivity analysis. We have added the relevant content in the "Introduction" section. Based on your suggestions, we have added two sensitivity analyses: "sensitivity simulation analysis of the probability \delta of the local government supervising shippers receiving subsidies" and "sensitivity simulation analysis of the penalty G_f imposed on shippers for noncompliant behavior in receiving subsidies if they fail to carry out the related freight transport as required.". Regarding your mention of "The government's investment subsidies for railway companies," we did not include it after further consideration, as the funding mechanisms for railway construction are highly complex, involving multiple stakeholders, diverse financing channels, and varying regional policies.

Action:

Please see page 34-37.

3. Article Structure: It is suggested to add some managerial implications after the conclusion to assist in the shift of cargo transportation from road to rail.

Response:

We appreciate your careful review of our manuscript. We have added the relevant content in the "Practical Recommendations" section of the "Inspiration and Discussion" chapter.

Action:

Please see page 41.

4. Grammar and Formatting: There are minor grammatical errors in the paper, and I recommend linguistic polishing to improve the readability of the manuscript.

Response:

We sincerely appreciate your feedback regarding language issues. Following the recommendations of the editorial team, we have utilized a professional language editing service.

Action:

Please refer to the appendix of this document.

5. Citation Formatting: There are citation formatting errors in Section 3.4 and Section 4.1. Please carefully proofread and correct these formatting issues.

Response:

We sincerely appreciate your help in correcting this error in our manuscript. We have made corrections in the relevant sections of "Stability Analysis of Equilibrium Points in the Tripartite Evolutionary Game System" under the "Model Solution and Analysis" chapter and "Parameter Assignment" under the "Simulation Analysis" chapter.

Action:

Please see page 22 and 29.

If you have any other questions or concerns, please feel free to contact the authors and we will respond and improve them seriously.

Reviewer #3:

The authors appreciate your statement on our work. We have revised the paper according to your comments. The green marked parts are the corresponding revisions.

1. The authors have selected a relevant and significant topic, presenting it clearly and demonstrating a solid grasp of the fundamental concepts. They have effectively highlighted the paper's contribution, and the technical rigor of the applied processes and statistical methods remains satisfactory. However, I noticed that the changes requested regarding the discussion section were not fully addressed. Specifically, I had recommended elaborating on your results by explaining how they align with or diverge from prior studies. Additionally, I suggested providing potential explanations for why your findings might support or contradict previous research and exploring their implications. Unfortunately, these aspects remain insufficiently developed. I strongly encourage revisiting the discussion section to incorporate these elements. Expanding on how your findings connect to existing literature will not only enhance the robustness of your manuscript but also provide readers with a deeper and more nuanced understanding of the topic. Addressing this will greatly strengthen the overall contribution and impact of your work.

Response:

We appreciate your careful review of our manuscript. Thank you for your valuable feedback and constructive suggestions. We greatly appreciate your thorough review and insightful recommendations, which have significantly helped us improve our manuscript.

Regarding your comments on the discussion section, we acknowledge that our initial revision did not fully elaborate on how our findings align with or diverge from prior studies, nor did it sufficiently explore potential explanations for these similarities and differences. In response, we have made substantial modifications to this section to strengthen the connection between our results and existing literature.

First, we have explicitly compared our findings with previous studies, particularly in terms of the participant selection in the evolutionary game model. Unlike prior research that included road transport enterprises or intermediary platforms, our study incorporates shippers as direct participants to better capture supply–demand dynamics in the transportation market. This distinction provides a more realistic depiction of decision-making processes and offers more targeted theoretical support for policy formulation and corporate strategy. In addition, we have expanded our discussion to highlight how the increasing policy support for railway transport in China aligns with global trends, as evidenced in studies on green transportation and sustainable development.

Second, we have explored potential reasons for the differences between our findings and previous research. Our results indicate that government subsidies play a crucial role in the early stages of road-to-rail transition, but their effectiveness gradually diminishes as market mechanisms take over. This observation aligns with findings on China Railway Express but differs in emphasizing the importance of well-designed subsidy exit strategies to ensure long-term market sustainability. Additionally, we discuss the observed asymmetry in the speed of strategy adjustments among stakeholders. Our findings suggest that shippers respond more quickly to policy incentives than railway transport enterprises, highlighting the necessity of synchronized policy interventions to ensure balanced progress across stakeholders. This dynamic perspective, which is often overlooked in previous static models, offers a more nuanced understanding of transportation structure adjustments.

Furthermore, we have expanded on the policy and strategic implications of our findings. We have provided a more detailed discussion on the role of government intervention in shaping market behavior and how targeted subsidies and regulatory measures can facilitate a smooth transition from government-driven to market-driven transport adjustments. At the same time, we have further clarified our study’s theoretical contributions, particularly in advancing evolutionary game modeling for transport policy analysis. Our framework captures the adaptive strategies of stakeholders over time, addressing an important research gap by integrating temporal dynamics into the study of transportation structure evolution.

We believe these revisions have significantly strengthened the discussion section, providing a clearer and more comprehensive interpretation of our findings. We sincerely appreciate your valuable suggestions, which have helped enhance the overall contribution and impact of our work.

Thank you again for your time and effort in reviewing our manuscript. We look forward to your further comments and suggestions.

Action:

Please see page 37-40.

If you have any other questions or concerns, please feel free to contact the authors and we will respond and improve them seriously.

---

## [Decision Letter · Decision Letter 2]

26 Feb 2025

Promoting the Modal Shift of Freight from Road to Rail in China An Evolutionary Game and Simulation Study

PONE-D-24-54352R2

Dear Dr. Liu,

We’re pleased to inform you that your manuscript has been judged scientifically suitable for publication and will be formally accepted for publication once it meets all outstanding technical requirements.

Kind regards,

Mohamed Rafik N. Qureshi, Ph.D.

Academic Editor

PLOS ONE

Additional Editor Comments (optional):

Thank you for the updated version.

Reviewers' comments:

Reviewer's Responses to Questions

**Comments to the Author**

1. If the authors have adequately addressed your comments raised in a previous round of review and you feel that this manuscript is now acceptable for publication, you may indicate that here to bypass the “Comments to the Author” section, enter your conflict of interest statement in the “Confidential to Editor” section, and submit your "Accept" recommendation.

Reviewer #1: All comments have been addressed

Reviewer #3: All comments have been addressed

2. Is the manuscript technically sound, and do the data support the conclusions?

Reviewer #1: Yes

Reviewer #3: (No Response)

3. Has the statistical analysis been performed appropriately and rigorously? 

Reviewer #1: N/A

Reviewer #3: (No Response)

4. Have the authors made all data underlying the findings in their manuscript fully available?

Reviewer #1: Yes

Reviewer #3: (No Response)

5. Is the manuscript presented in an intelligible fashion and written in standard English?

Reviewer #1: Yes

Reviewer #3: (No Response)

6. Review Comments to the Author

Reviewer #1: The authors' attempt is a difficult one and obtaining a valuable working tool for transport management is obliged to use the created method over time and improve it. The detection of neglected aspects and the increase in prediction accuracy is linked to the permanent use of the method in railway and road transport monitoring operations. The application of the method must be done on real data and in several geographical areas. I hope that the authors will continue their efforts.

Reviewer #3: Thank you for updating the manuscript. I truly appreciate your efforts and the hard work you have put into it. I have no further comments.

7. PLOS authors have the option to publish the peer review history of their article (what does this mean? ). If published, this will include your full peer review and any attached files.

**Do you want your identity to be public for this peer review?** For information about this choice, including consent withdrawal, please see our Privacy Policy .

Reviewer #1: **Yes: ** Petru Cardei

Reviewer #3: No

---

## [Editor Report · Acceptance letter]

PONE-D-24-54352R2

PLOS ONE

Dear Dr. Liu,

I'm pleased to inform you that your manuscript has been deemed suitable for publication in PLOS ONE. Congratulations! Your manuscript is now being handed over to our production team.

Kind regards,

on behalf of

Prof.(Dr.) Mohamed Rafik N. Qureshi

Academic Editor

PLOS ONE